

# Polarimetric radar characteristics of lightning initiation and propagating channels

Jordi Figueras i Ventura[1], Nicolau Pineda[2,3], Nikola Besic[1], Jacopo Grazioli[1], Alessandro Hering[1], Oscar A. van der Velde[3], David Romero[3], Antonio Sunjerga[4], Amirhossein Mostajabi[4], Mohammad Azadifar[5], Marcos Rubinstein[5], Joan Montanyà[3], Urs Germann[1], and Farhad Rachidi-Haeri[4]

[1]MeteoSwiss, Locarno, Switzerland
[2]Meteorological Service of Catalonia, Barcelona, Spain
[3]Universitat Politècnica de Catalunya UPC, Terrassa, Spain
[4]École Polytechnique Fédérale de Lausanne EPFL, Lausanne, Switzerland
[5]Haute Ecole Spécialisée de Suisse occidentale HES-SO, Yverdon-les-bains, Switzerland

**Correspondence:** Jordi Figueras i Ventura (jordi.figuerasiventura@meteoswiss.ch)

**Abstract.** In this paper we present an analysis of a large dataset of lightning and polarimetric weather radar data collected in the course of a lightning measurement campaign that took place in the summer of 2017 in the area surrounding the Säntis mountain, in the northeastern part of Switzerland. For this campaign, and for the first time in the Alps, a lightning mapping array (LMA) was deployed. The main objective of the campaign was to study the atmospheric conditions leading to lightning production

5     with particular focus on the lightning discharges generated due to the presence of the 124 m tall Säntis telecommunications tower. In this paper we relate LMA VHF sources data with co-located radar data in order to characterize the main features (location, timing, polarimetric signatures, etc.) of both the flash's origin and its propagation path. We provide this type of analysis first for the whole data and then we separate the datasets into intra-cloud and cloud-to-ground flashes (and within this category positive and negative flashes) and also upward lightning. We show that polarimetric weather radar data can be helpful

10     in determining regions were lightning is more likely to occur but that lightning climatology and/or knowledge of the orography and man-made structures is also relevant.

*Copyright statement.* TEXT

## 1 Introduction

The first LMAs were introduced in South Africa by D.E. Proctor (Proctor, 1971, 1981; Proctor et al., 1988). They have

15     gradually experienced significant improvements both in terms of software and hardware and in the understanding of their limitations (e.g. Thomas et al., 2004; Fuchs et al., 2016) and they have become a fundamental tool in the lightning research community. Networks have been deployed or are currently operational in various locations in the USA and Europe either on a permanent basis (e.g. Koshak et al., 2004; van der Velde and Montanyà, 2013) or in the context of specific measurement campaigns (e.g. Defer et al., 2015).



Coupling LMA data with information obtained from polarimetric weather radar, one can obtain valuable information about the type of precipitating system that produced the lightning activity and the sort of environment where the flashes propagate. In particular, polarimetric weather radars can inform of the dominant hydrometeor type in the region where the flash propagates. Numerous studies using both LMA and polarimetric weather radar data have been presented in the past, although most of them

were discussing data from individual storms or types of storms. For example, in the context of the Thunderstorm Electrification and Lightning Experiment TELEX measurement campaign, in the US planes, (MacGorman et al., 2008), several case studies were published about for example observations of a multicell storm (Bruning et al., 2007), a small mesoscale convective system (Lund et al., 2009) or of the presence of a lightning ring in a supercell storm (Payne et al., 2010). Within the context of another measurement campaign in the US planes, the Severe Thunderstorm Electrification and Precipitation Study STEPS (Lang et al.,

2004), observations were presented about storms with normal and inverted polarity (Tessendorf et al., 2007). More recently, Kumjian and Deierling (2015) examined lightning-producing snow storms.

Some well-known polarimetric signatures of electrification processes, have been discussed in the literature in the course of the years. For example, it is widely reported that strong electric fields may align ice particles resulting in negative values of the differential reflectivity $Z_{dr}$ and specific differential phase $K_{dp}$ (e.g. Figueras i Ventura et al., 2013; Hubbert et al., 2014).

Other authors have observed that the presence of a $Z_{dr}$ column is an indicator of a strong updraft (Snyder et al., 2015), which has been repeatedly reported to favor lightning activity (Calhoun et al., 2013). Hence the presence of a $Z_{dr}$ column would be an indirect indicator of possible lightning activity. There is also considerable evidence that the presence of large quantities of graupel (retrievable with polarimetric weather radars) in an environment with supercooled water leads to the production of lightning through noninductive charging (e.g. Saunders et al., 2006; Ribaud et al., 2016).

Other studies presented relevant features of lightning activity extracted from large datasets of LMA data but they did not provide a direct connection with the precipitation regime as observed by polarimetric radars. For example, López et al. (2017), examined the spatio-temporal characteristics of 29000 flashes in the Norht-East of the Iberian Peninsula. Chronis et al. (2015) used a large dataset obtained by the Sao Paolo LMA to construct a climatology of the diurnal variability of lightning activity in the area. Fuchs et al. (2015) did use a large dataset of storms data from 4 different LMA networks in the USA and the

3D mosaic of reflectivity from the WSR88D network to determine the environmental characteristics affecting the electrical behavior of storms. The authors focused essentially on the vertical profile of reflectivity of the convective storms. Another notable study from Mattos et al. (2016), establishes relationships between the vertical profiles of polarimetric data obtained from an X-band Doppler polarimetric weather radar and the flash density.

This paper reports on a measurement campaign that took place in the area around the Säntis mountain, in the northeastern

part of Switzerland during the summer of 2017. Within this campaign, and for the first time in the Alps, a Lightning Mapping Array (LMA) was deployed. The main objective of the campaign was to study the atmospheric conditions leading to lightning discharges generated due to the presence of the 124 m tall Säntis telecommunications tower. In the course of the campaign, a sufficient amount of data was collected to allow a more general analysis of the atmospheric conditions leading to lightning generation and propagation. In this respect, LMAs offer a unique dataset since they provide 3D information of the discharge

path, including channels within the cloud, with acceptable temporal and spatial precision.



In this study we use a relatively large dataset of LMA data (more than 12000 flashes collected over 8 different days) and a nearby operational C-band Doppler polarimetric weather radar to determine the most likely precipitation conditions for both flash generation and flash propagation. The study makes use of the basic polarimetric variables as well as the derived hydrometeor classification. Moreover, and for the first time, we go one step further and we apply a technique that allows us

to obtain semi-quantitative information on the composition of the precipitating system at sub-radar resolution volume scale. We will provide a general analysis of all the LMA data collected and proceed to its categorization according to whether they produced cloud-to-ground (CG) lightning or not (and according to the CG flash polarity) and for upward lightning.

LMAs require direct line-of-sight to operate and therefore, as a general rule, they do not provide information from the lowest layer of the atmosphere. Hence they are not capable of directly sensing CG flashes. Indirect information of flashes reaching the

ground can be obtained by observing the presence of stepped leaders, i.e. sets of VHF sources propagating downwards. Dart leaders, on the other hand, are poorly traced because they propagate more directly to the ground. Networks of low frequency sensors, such as the EUCLID network in Europe, on the other hand, offer a high probability of detection of CG lightning data with good location accuracies. Their detection capability of intra-cloud (IC) lightning, however, is much more limited. It is therefore clear that the data offered by both sensors are largely complementary. In this paper we have used data from both

networks to distinguish those flashes detected by the LMA that have produced lightning strokes to the ground according to the EUCLID network from those which have not. Moreover, since the EUCLID network provides information about the polarity of the lightning stroke, we can further distinguish the LMA flashes causing positive CG strokes and those causing negative CG strokes.

Upward lightning is a source of continuous interest in the research community. This type of lightning is generally associated

with the existence of tall structures and, with the global expansion of wind farms, its incidence is likely to increase in the future (Rachidi et al., 2008). However, a significant number of this type of flashes is not detected by conventional lightning detection systems since they might contain only an initial continuous current with neither superimposed pulses nor return strokes (e.g. Diendorfer et al., 2009; Smorgonskiy et al., 2013; Azadifar et al., 2015). Moreover, another significant portion is actually detected but misclassified as IC. In this study, we have collected all flashes detected by the LMA with the first VHF source

located in the liquid or mixed phase regions of the precipitating system according to the radar-based hydrometeor classification. Although detectability issues may not be ruled out, especially in such complex orography, the flashes thus collected are more likely to correspond to genuine upward lightning and hence they deserve a separate analysis of their propagation conditions.

Summarizing, the main goals of this study are:

- To determine whether and which polarimetric signatures can inform us of the likelihood of ligthning activity.

- To find out whether polarimetric signatures can be used to distinguish between precipitating conditions producing mostly IC lightning activity from those producing CG activity.

- To study from a statistical perspective the environmental conditions leading to upward lightning.

The paper is organized as follows: Section 2 provides an overview of the Säntis measurement campaign, the instrumentation used and the data processing methods. Section 3 contains a detailed data analysis. Firstly, the data availability and coverage is

discussed, followed by a general overview of all the data analyzed and finally the data are classified in different categories: intra-



cloud, cloud-to-ground (positive or negative) and flashes with origin in the mixed phase or liquid layers. General conclusions
and recommendations are discussed in section 4.

## 2 The Säntis Measurement Campaign

The Säntis measurement campaign was a joint venture between the Electromagnetic Compatibility Laboratory (EMC LAB) of
the Swiss Federal Institute of Technology in Lausanne (EPFL), the Institute for Information and Communication Technologies
of the University of Applied Sciences of Western Switzerland (HEIG-VD ), the Lightning Research Group (LRG) of the
Technical University of Catalonia, the Meteorological Service of Catalonia (Meteo.cat) and the Radar Satellite and Nowcasting
Division of the Federal Office of Meteorology and Climatology MeteoSwiss. The campaign took place in the summer of 2017.
The main objective of the campaign was to study the atmospheric conditions leading to lightning production in the vicinity of
the Säntis telecommunications tower, with particular focus on the upward lightning discharges generated by the tower itself.
The 124 m tall telecommunications tower is situated on top of the Säntis mountain (47.2429°N, 9.3393°E, 2502 m MSL), in the
Sankt Gallen Canton, in the north-east of Switzerland. The main instruments of the campaign were in-situ measurements on the
tower, a Lightning Mapping Array (LMA) network and a polarimetric Doppler weather radar. The area covered by the campaign
and the location of the instrumentation can be seen in Fig. 1. In the following, a brief description of the instrumentation used
during the campaign is provided.

### 2.1 Ligthning mesurements

#### 2.1.1 Ligthning measurements at the Säntis tower

Since May 2010, EPFL and HEIG-VD operate instrumentation to detect and characterize lightning strikes on the Säntis tower.
Lightning currents in the tower are measured using two sets of Rogowski coils and multigap B-dot sensors located at two
different heights along the tower (82 m and 24 m, respectively) (Romero et al., 2012). The analog ouputs are relayed to a
digitizing system by means of optical fiber links. A PXI platform digitizes and records the measured waveforms at a sampling
rate of $50 \, \mathrm{MS \cdot s^{-1}}$. The ligthning current is recorded over a 2.4-s time with a pre-trigger delay of 960 ms. The system is GPS
equipped and allows remote maintenance, monitoring and control via Internet.

Since 15 July 2016, an EFM-100 field mill is installed 85 m from the tower to measure the electrostatic field. The system can
detect lightning activity up to distances of about 40 km from the tower (Azadifar, 2017). In addition to that, an electric field
measuring system comprising a flat plate antenna and an analog integrator with an overall operating frequency band between
30 Hz and 2 MHz is installed 14.7 km away from the Säntis tower (He et al., 2018).

#### 2.1.2 Lightning Mapping Array

LRG owns and operates a 3D LMA installed on a permanent basis at the lower part of the Ebro Valley, in Catalonia. The
LMA network there consists of 12 VHF (60-66 MHz) sensors, some of which are mobile. For this campaign, 6 sensors were





moved temporarily to locations in the area surrounding the Säntis (See Table 1 and Fig. 1 for the locations). The network was operational from June 29 to August 15 2017, although not all the sensors were operating during the entire period. The selection of the locations was made taking into consideration practical installation aspects such as accessibility, security and reliable access to AC power and communication as well as considerations over the magnitude of the local noise within the frequency

band and the distance to the Säntis tower. In the end, the sensors were located in the vicinity of mobile phone base stations belonging to Swisscom and Swisscom Broadcast, which in the cases of the Gonten, Schwägalp and Säntis stations resulted in an increased noise level coming from the on-site telecommunications equipment. The Säntis station had the added challenge of being located indoors.

Each sensor measures the arrival times of the impulsive VHF radiation sources with an accuracy of 50 ns using a PC-based

digitizer card coupled to a GPS receiver. The received signal is digitized and the peaks (within a 80 µs window) are time-tagged using the time derived from GPS receivers, which provide a timing pulse once per second (Rison et al., 1999). The timed data are stored on-site as well as transmitted over wireless modems to a central site for real time analysis and display. If at least 4 stations are able to measure the time of arrival of the radiation from a particular impulsive event, the 3-dimensional position of the source region can be estimated since this is described by 4 unknowns (the three position coordinates $x$, $y$, $z$ plus the exact

time when the discharge occurred $t$). A redundancy in the number of stations observing the event allows for a better accuracy and helps filtering out noise spikes from other source types. After post-processing, individual sources deemed to be part of the same flash are clustered together and assigned a unique ID number (Fuchs et al., 2015). If the data point is not identified as belonging to any flash, it is assigned the ID number 0. A file with data from all the sources is generated each day. In this campaign, in order to account for the increased noisiness of the data and the poor visibility, a minimum of 10 detected VHF

sources is required for a flash to be accepted.

Since the detection of the lightning strike requires direct line of sight of the source, LMAs observe mainly IC activity, mostly from negative leaders moving through regions of positive charge. However, weaker sources from positive leaders moving through negative charge regions are often detected. CG activity is often detected indirectly from stepped negative leaders or, less often, from negative dart leaders and some times positive leaders as well.

### 2.1.3 EUCLID lightning detection network

Operational detection of CG and IC activity is performed by the European Cooperation for Lightning Detection Network (EU-CLID) (Schulz et al., 2016). The network uses various Vaisala sensors that detect low frequency (1-350 kHz) electromagnetic signals and GPS receivers providing the time. The raw data are sent to a centralized location where they are processed and for each lightning strike the time of event, latitude and longitude of the impact point and the current intensity and polarity are

recorded. The EUCLID network has a high CG detection efficiency (on the order of 95%), but a reduced IC detection efficiency, typically on the order of 50%. The median accuracy is on the order of 100 m although it may be worse in mountainous areas (Azadifar et al., 2015). The data available at MeteoSwiss is provided by Météorage and has a time resolution of 0.1 s.



## 2.2 Polarimetric weather radar data

MeteoSwiss owns and operates a network of 5 C-band, Doppler polarimetric weather radars. The network was recently renewed within the project Rad4Alp, which was concluded in 2016 (Germann et al., 2015). The 5 systems have identical specifications and modes of operation. The scanning strategy consists of 20 horizontal scans with elevations ranging from -0.2° to 40°

repeated every 5 min. The elevations are inter-leaved: every 2.5 min a half-volume of 10 elevations from top to bottom is concluded. A very short pulse of 0.5 μs is used to obtain data with a range resolution of 83.3 m with angular resolution of 1°. IQ data are processed on-site using standard techniques (e.g., Doviak and Zrnic, 2006) to obtain the basic polarimetric moments, i.e., reflectivity (horizontal $Z_h$ and vertical $Z_v$), differential reflectivity ($Z_{dr}$), co-polar correlation coefficient ($\rho_{hv}$) and raw co-polar differential phase ($\psi_{dp}$) as well as Doppler moments. These basic moments are transmitted to a central server. The

operational data processing involves a clutter detection using a sophisticated decision tree filter (DT-filter) and a reduction of the resolution to 500 m by averaging 6 consecutive gates (only clutter-free ones). From the low resolution polarimetric moments all subsequent products are generated. For the measurement campaign, data from the Albis radar (47.2843°N 8.5120°E, 938 m MSL), situated 63 km west of the Säntis tower (see Fig. 1) were used.

### 2.2.1 Basic radar data processing

A specific non-operational processing was applied to radar data obtained in real time during the campaign. The processing was performed using the Python-based open source software Pyrad/Py-ART (Figueras i Ventura et al., 2017). The first step was calculating the signal-to-noise ratio (*SNR*) of the horizontal channel using the estimated receiver noise from a high elevation (40° or 35°) angle. The *SNR*, together with the ratio between the horizontal and vertical channels receiver noise were used to minimize the effect of noise on $\rho_{hv}$ (Gourley et al., 2006). Clutter identification was performed using a simple DT-filter based

on the textures of $Z_h$, $Z_{dr}$, $\rho_{hv}$ and $\psi_{dp}$ and the value of $\rho_{hv}$. Range gates identified as clutter-contaminated were removed from the analysis. $\psi_{dp}$ was processed by first filtering out range gates with *SNR* below 10 dB to reduce the influence of phase noise and then applying a double window moving median filter. The length of the windows was 1000 and 3000 m, respectively. The short window was applied in regions of high reflectivity (above 40 dBZ) while the long window was used elsewhere. The filtered differential phase ($\phi_{dp}$) was used to compute the specific attenuation $A_h$ using the ZPhi algorithm (Ryzhkov et al., 2014).

$A_h$ was estimated up to the freezing level height as determined by the temperature provided by the closest available run of the Numerical Weather Prediction (NWP) model COSMO-1 (see http://www.cosmo-model.org/). From $A_h$, the specific differential attenuation $A_{dp}$ was derived. By integrating $A_h$ ($A_{dp}$) attenuation, the Path Integrated (Differential) Attenuation was obtained and this quantity was added to the (differential) reflectivity in order to correct for the precipitation-induced attenuation. In parallel to the attenuation correction, the specific differential phase ($K_{dp}$) was derived from $\phi_{dp}$ using the method described in

Vulpiani et al. (2012).

$K_{dp}$, $Z_h$, $Z_{dr}$, $\rho_{hv}$ and the temperature from the COSMO model were inputs of the semi-supervised hydrometeor classification described by Besic et al. (2016). The hydrometeor classification provides the following outputs: aggregates (AG), ice crystals (CR), light rain (LR), rimed particles (RP), rain (RN), vertically-oriented ice crystals (VI), wet snow (WS), melting hail (MH),




ice hail-high density graupel (IH) and no classification (No valid radar data) NC. It must be mentioned here that the category vertically-oriented ice crystals is often misclassified. It is highly dependent on the $Z_{dr}$ value, with the assumption of $Z_{dr}$ being negative when the particles are vertically-oriented. Unfortunately, at C-band large values of $A_{dp}$ are not uncommon. Although $A_{dp}$ is corrected for, insufficient correction typically results in negative $Z_{dr}$ values at the far-end of the ray. Therefore the

category vertically-oriented ice crystals may contain a large proportion of regularly-oriented ice crystals situated at range gates far away from the radar.

In addition to the dominant hydrometeor class, the MeteoSwiss hydrometeor classification algorithm also provides an estimation of the entropy (Besic et al., 2018). Such quantity provides information about whether within the radar resolution volume there is a clearly dominant hydrometeor type (entropy 1) or if it is an heterogeneous mixture without any dominant

hydrometeor type (entropy 0). Moreover, using the technique described in the aforementioned Besic et al. (2018), we are able to extract semi-quantitative information of the proportion of each hydrometeor type contained in the resolution volume.

At the end of the processing, high resolution clutter-free volumes of attenuation-corrected $Z_h$ and $Z_{dr}$, $\rho_{hv}$, $K_{dp}$, the model air temperature, the dominant hydrometeor type, the entropy and the proportion of each hydrometeor type at each range gate were obtained. These parameters were used in the subsequent analysis.

### 2.2.2  Radar data along lightning trajectories

Within the radar data processing tool Pyrad, a lightning trajectory function has been implemented. This function reads the daily produced LMA lightning data and determines from them the time of the first and last VHF source detection. It then loops over all the radar volumes within this time interval and assigns to each VHF source the value of the polarimetric parameter in the range gate closest (both in time and space) to the source location. For each radar variable a file is generated containing the flash

source time, the flash number, the value at flash position and the mean, minimum and maximum of the cube formed by the neighbors.

### 2.2.3  Association of LMA flashes to EUCLID CG strokes

A two-step algorithm associates LMA flashes to corresponding EUCLID CG strokes. The first step looks for EUCLID CG strokes occurring during the propagation time of each LMA flash. A 0.1 s tolerance is added to the start and the end of the

LMA flash since this is the time resolution of the EUCLID data available at MeteoSwiss. If EUCLID strokes have been found within the LMA flash duration, a second step looks whether these EUCLID strokes are within the area covered by the LMA flash. The LMA flash area is defined as the minimum oriented rectangle that contains all the VHF sources of the flash projected to the ground. The minimum area of this rectangle is set to be 25 km$^2$, which is considered a reasonable area to look for in case of flashes with reduced horizontal extend. A scaling factor of 1.2 is applied to each dimension of the rectangle if its

area is larger than the minimum area to account for the fact that the flash may still propagate horizontally below the LMA detectable altitude. If EUCLID strokes are present within the LMA flash area the LMA flash data are saved in a separate file. This process can be performed to associate LMA flashes to all, positive or negative CG EUCLID strokes. A separate process



gets the complementary data, i.e. LMA flashes without associated CG EUCLID strokes, by comparing the file with all the flash data with the resultant filtered data file.

## 3 Data analysis

### 3.1 Data availability and coverage

The LMA was installed in the Säntis area between 29 June and 15 August 2017. On half of the days that the campaign lasted (24 out of 48) some lightning activity was registered in the area covered by the LMA by the EUCLID network. Of these, on 15 days lightning activity was registered within 2 km from the Säntis tower. On 10 of these, direct strikes to the tower were registered also by the in-situ sensors. According to the operational MeteoSwiss Probability of Hail (POH) algorithm, hail with a probability above 90% was present somewhere within the domain on 11 days. On 6 of these days hail was detected

within 20 km of the Säntis tower. Fig. 2 provides an overview of the relevant events during the campaign. For our analysis, we have focused on events that produced lightning in the immediate proximity of the tower. Of these, 22, 24 and 25 July were excluded because less than 5 LMA stations where operational. The 5, 8, 9 and 15 of August were excluded as well because although enough stations were operating, for reasons still under investigation the data quality was poor. Some characteristics of the events with lightning in the vicinity of the Säntis tower can be seen in Table 2. During the 8 days where LMA data were

analyzed, a total of 1586394 VHF sources, corresponding to 12062 flashes, were detected (i.e. 132 VHF sources per flash). Almost half of them were detected on the first of August alone.

Fig. 3 top left plots the position of all the LMA detected VHF sources during the days analyzed. Each cross in the plot is a detected LMA VHF source. The data have been color-coded by estimated altitude with lowest altitude in dark blue. The lowest altitude VHF sources are plotted on top. As the plot shows, the data are distributed over a broad surface oriented from south-

west to north-east, which was the moving direction of most of the convective cells during the period analyzed. It is also apparent that the coverage of the LMA network is uneven. There is a section in the south, oriented from south-west to north-east, where the minimum altitude at which data are detected is much higher due to blockage from the Alps. Likewise, there is a gradual loss of detectability the further one moves away from the center of the network. Since the number of operational sensors varied between 5 and 6 during the days examined and the malfunctioning sensor was not always the same, the actual detectability

varies between days. Nevertheless, the area surrounding the Säntis mountain appears to be reasonably well covered. Fig. 3 top right plots the position of the first LMA detected VHF source of each flash during the days analyzed. It can be seen that they have an uneven distribution with a band with high density of flashes to the south and a second band with a lower flash density further north. However this seems to be associated more to the path of individual storms than to specific detectability issues.

### 3.2 General data analysis

This section discusses the bluish histograms in Fig. 4 to Fig. 8 that provide a general overview of all the data sources collected during the days analyzed. It should be noticed that in all histograms presented in this paper, the values outside of the histogram





range are added to the bins at the extremes, i.e., the last bin in the histograms at Fig. 4 top left include all the values above 900 ms.

Concerning the distribution of the flashes over the day (see Fig. 4 top left panel), they are concentrated in the afternoon. The first flashes are detected at 10 UTC and the last at 21 UTC with a clear peak at 16 UTC. Obviously, with so few events, the

distribution is highly dependent on the severity and time of occurrence of the individual events but it seems probable that in the study area there is a diurnal cycle with a peak at mid-afternoon during the summer months. Fig. 4 top right panel shows the distribution of the duration of the flashes. For our purposes we define the flash duration as the time difference between the first and the last VHF source identified as belonging to a single flash. As it can be seen it has an exponential distribution. 51% of flashes have a duration of up to 200 ms. On the other hand, 3% of flashes have a duration of more than 900 ms. The flash area

(see Fig. 4 bottom panel) also follows a marked exponential distribution with more than half of them (58%) covering an area of less than 100 $km^2$. There are few flashes though (0.4%) that cover an area of more than 1900 $km^2$.

Fig. 5 shows histograms of the power of the VHF sources detected by the LMA. On the left panel all sources are shown while on the right only the first detected source for each flash (which we consider a proxy for flash origin) is shown. The histogram exhibits a Gaussian-like shape with median of 18.5 dBm. Source powers detected range from -16 to 46 dBm. The

sources power at the origin has also a Gaussian-like shape but with increased power. The median in this case is 20.5 dBm.

Fig. 6 top panels show histograms of the altitude of the VHF sources detected by the LMA. It can be seen that whereas the altitude histogram of the flashes origin has a Gaussian-like shape with mode 8600 m MSL and median 8200 m MSL (Fig. 6 top right), when all sources are considered the histogram has a bi-modal shape with two distinct peaks, the main one at 7400 m MSL and a secondary one at 4000 m MSL ((Fig. 6 top left). This suggests that most of the IC lightning activity is generated

at the higher part of the clouds with the majority of the flashes propagating roughly horizontally but a significant proportion descending into lower layers. This observation is further confirmed by Fig. 6 bottom panels in which the temperature at the location of each lightning source, obtained from the COSMO-1 NWP model, can be seen. Indeed, the vast majority of the sources are detected at freezing temperatures. The median temperature for all sources is -20°C while the lightning origin is at -25°C.

The reflectivity data (topmost panels in Fig. 7) of all sources show a bi-modal distribution with a main peak at 41 dBZ and a secondary peak at 25 dBZ. Interestingly, when looking only at the first source the main peak is maintained (at 41.5 dBZ), but the secondary peak is barely visible. The $Z_{dr}$ data (upper-middle panels) exhibit a similar Gaussian-like shape both when all sources are considered and when only the first source is considered. In both cases the distribution is centered around 0 but with very long tails. $\rho_{hv}$ data (lower-middle panels) have a log-normal shape with most values well-above 0.99. The mode is

0.9990 both when considering all sources and when only considering the first one. $K_{dp}$ values (lowermost panels) are centered at $0°km^{-1}$ but they are noticeably skewed towards positive values.

Fig. 8 top panels show histograms of the dominant hydrometeor at the location of the VHF sources. As it can be seen, the large majority of the sources are produced in areas of rimed particles (54.58%) or dry hail (26.85%). When considering only first sources, the proportions of these species are essentially maintained, with a slight increase of the solid hail proportion

(54.18% for rimed particles and 29.52% for solid hail). The other classes with significant VHF sources are dry snow (11.73%)



and ice crystals (1.68%, of which 71.61% correspond to vertically-aligned ice crystals, which is indicative that those are ice crystals situated at the far end of the radar, i.e. high up in the atmosphere). Again, similar proportions are encountered when examining only first sources (9.26% for dry snow and 1.96% for ice crystals, of which 63.14% vertically-aligned). The lower proportion of vertically-aligned ice crystals may indicate that flashes are less likely to originate at the very top of the cloud.

1.78% of the VHF sources transit through the wet snow area but only 0.83% of the flashes are generated there. 1.60% of the sources are detected in the rain medium (of which only 22.51% in areas of light rain) but only 0.93% (33.93% of which in light rain) of the flashes are generated there. 0.35% of the flashes transit through areas were hail in the process to be melted is present (0.24% of the flashes originate in such region). Summarizing, 94.92% of the flashes detected originated at the solid phase region of the precipitating systems whereas only 2.00% originated in the mixed-phase or liquid regions of the precipitating systems.

3.07% have origin in regions where no radar echo was detected but examining the location (not shown) it can be inferred that for the most part they correspond to the solid phase region.

Fig. 8 medium panels show histograms of the entropy of the hydrometeor classification at the location of the VHF sources. It is evident from the graph that the entropy tends to be rather high with values on the order of 0.3 and 0.4 dominating the distribution and representing 62.4% of all the VHF sources and 64% of all the flash origins where radar data were present.

Indeed, when looking at how many hydrometeors types have a significant presence within the radar gates colocated with VHF sources (that is more than 10% contribution to the hydrometeor proportions) (see Fig 8 lower panels), it turns out that in a large majority of them there is more than one hydrometeor type and up to a maximum of 6. 69% are composed of 2 hydrometeor types while only 22% contain one single dominant hydrometeor. If looking at the flash origin only, the number of gates containing only 1 hydrometeor type decreases further by one point (21%).

Fig. 9 top panels show 2D-histograms of the most-dominant and second-most dominant hydrometeors. The most likely combination of hydrometeors in the presence of VHF sources by far is a combination of rimed particles as dominant type and solid hail as second most common. The second most likely is the combination of solid hail as most dominant and rimed particles as second most common. The third is a combination of rimed particles as dominant and aggregates as second most common while the fourth is rimed particles as single dominant hydrometeor. When focusing on the flash origin this ranking

is essentially maintained but the likelihood of a combination of rimed particles and aggregates decreases. It seems clear from these results that lightning tends to originate in areas with an important presence of rimed particles or solid hail.

### 3.3 Characteristics of the flashes without associated EUCLID cloud-to-ground strokes

We analyze here the characteristics of flashes without associated EUCLID CG strokes, which we use as a proxy for IC flashes. This section discusses the greenish histograms in Fig. 4 to Fig. 8. 90.30% of the total flashes (10892) correspond to this cate-

gory. Those flashes contain 81.94% of the total detected sources (1299821). The fact that flashes without associated EUCLID CG strokes have less sources per flash (119 respect to 132) may be an indicator that this type of flashes are more short-lived and with simpler structures.

Since a very large proportion of flashes are IC flashes, their characteristics do not differ significantly from the general analysis performed in the previous subsection. The VHF source power distribution (see Fig. 5) is very similar. The VHF source



altitude (Fig. 6) has a more marked bi-modality as well as the reflectivity distribution (Fig. 7 top panels) when all sources are considered.

The distribution of the values of the hydrometeor classification is the following (see Fig. 8 top panels): When all sources are considered, 55.09% can be found in areas with rimed particles, 25.89% in areas with solid hail, 11.96% in dry snow and 5 1.85% in ice crystals areas (of which 70.05% are vertically oriented), 1.76% in wet snow, 1.50% in rain (of which 24.28% in light rain) and 0.32% in melting hail. 1.62% propagate in areas where no radar echo was detected. When looking only at the first sources, 54.85% are produced in areas with rimed particles, 28.38% in areas with solid hail, 9.49% in dry snow, 2.05% in ice crystals areas (of which 62.61% are vertically oriented), 0.79% in wet snow, 0.90% in rain (37.76% of which in light rain) and 0.24% in melting hail. 3.29% originate in areas where no echo was detected. Compared to the global data, it is worth 10 noticing the lower percentage of flashes originating in areas where solid hail is predominant. As it is the case with all the data, the entropy is quite high (see Fig. 8 medium panels) and most of the radar gates where VHF sources are located contain more than one hydrometeor type (see Fig. 8 bottom panels). Rimed particles or solid hailstones are present in most regions where the flashes propagate (see Fig. 9 medium panels).

### 3.4 Characteristics of the flashes with associated EUCLID cloud-to-ground strokes

15 We analyze here the characteristics of flashes that have associated EUCLID CG strokes, hereby CG flashes. This section discusses the reddish histograms in Fig. 4 to Fig. 8. This category contains a total of 1085 flashes which correspond to 9.0% of the total. Those flashes contain 17.60% of all the sources detected during the days analyzed (279199). On average the CG flashes during the campaign contain 257 sources. This would suggest that those flashes have a more complex structure than the IC flashes. Fig. 3 (bottom panels) shows the spatial distribution of all the VHF sources of these flashes (left) and the first VHF 20 source of each flash (right). The spatial distribution of all VHF sources approximately covers the same area that was covered by all data. However the density of the flashes origin is much lower and there are few detections in the northest band of lightning activity, suggesting that individual storms have a varying ratio of CG to IC flashes. That is further confirmed by looking at the time of occurrence of the flashes (see Fig. 4 top left panel), since the shape of the distribution changes significantly. The flash duration distribution (Fig. 4 top right panel) does not follow an exponential distribution anymore but has a mode of 350 ms and 25 a large number of flashes (17%) have a duration of more than 900 ms. The area covered by the flashes (Fig. 4 bottom panel) tends to be also larger. Only 15% of the flashes cover an area of less than 200 km$^2$ and 3% of the flashes cover an area of more than 1900 km$^2$.

In terms of source power (Fig. 5), CG flashes exhibit similar characteristics to those of the global analysis. Their histogram has again a Gaussian-like shape with the median a bit higher (19.5 dBm). When only first sources are considered the median is 30 the same as in the global analysis, 20.5 dBm. Regarding the altitude of the VHF sources, there are remarkable differences (see Fig. 6 top panels) with the global data. When all sources are considered, the histogram exhibits a Gaussian-like distribution but skewed towards lower altitudes with a median of 7000 m MSL and a mode of 7400 m MSL. When only considering the origin of the flashes, the histogram has an almost uniform-like distribution with mode 8700 m MSL and median 7900 m MSL. There is a 300 m difference between the median of CG flashes with respect to the global data which may indicate that those flashes





are more likely to be generated at lower altitude. This is backed also by the fact that the median temperature at flash origin (Fig. 6 botom right panel) is -25°C compared to the -28°C of the global data.

In terms of distribution, the polarimetric variables show subtle differences with respect to the global data (see Fig. 7). When all sources are considered, the secondary peak in the reflectivity distribution is almost unnoticeable and the mode is lower

(39 dBZ) and the median higher, 36.5 dBZ. When only the origin is taken into account, both the median and the mode are decisively larger (40 dBZ and 44 dBZ respectively). The $Z_{dr}$ distribution is similar but with comparatively fatter tails. $\rho_{hv}$ has slightly lower median (0.996 both when all sources are considered and when only the first source is considered). The $K_{dp}$ distribution is very similar but shifted towards higher values, particularly for the source origin, where the median has moved to $0.15\,°\mathrm{km}^{-1}$.

Although the general distribution is maintained with respect to the global data analysis, the proportions of each hydrometeor class are slightly different (see Fig. 8 top panels). When considering all VHF sources in the solid phase, their proportion decreases slightly but not dramatically (52.18% for rimed particles, 31.26% for solid hail, 10.63% for dry snow and 0.95% for ice crystals of which 86.03% are vertically oriented). As would be expected, the percentage of sources transiting through the wet snow region and the rain medium is remarkably higher (1.90% and 2.08% respectively) while 0.49% of the sources

were detected in areas of melting hail. 0.51% of the sources where located in areas with no radar echo. When looking at the origin of the flashes, the percentages of flashes originating in the solid phase decreases with respect to the global data (47.18% for rimed particles, 41.37% for solid hail, 7.02% for dry snow, 0.83% of ice crystals, of which 66.67% are vertically aligned). It is interesting to notice that there is a marked increase in flashes having their origin in solid hail regions and a decrease of flashes produced in the dry snow or ice crystals regions with respect to the global data. Flashes generated in the wet snow area

constitute 1.29% of the total while another 1.29% are generated in the rain medium. Finally, 0.28% of the flashes are generated in areas of melting hail. The percentage of VHF sources where hydrometeor classification could not be performed is lower than in the global analysis (0.51% when considering all flashes and 0.74% when considering flash origins), which corroborates the statement that non classified data are located mostly at high altitude.

As was the case with the global data, the entropy of the hydrometeor classification is rather high, with a large percentage

of VHF sources located in regions with entropy in the order of 0.3-0.4 (see Fig. 8 medium panels). Again, flashes are more likely to be generated and propagate in areas where at least 2 hydrometeor types are present in significant proportions (see Fig. 8 lower panels).

Fig 9 bottom panels show 2D-histograms of the most-dominant and second-most dominant hydrometeors for sources associated with EUCLID CG strokes. When looking at all sources (Fig. 9 bottom left panel) the distribution is rather similar to the

global data. The main difference is that the combination of solid hail as dominant and rimed particles as second most dominant has a comparatively higher weight. However, when looking at the flash origin (Fig. 9 bottom right panel) the mentioned combination becomes the most likely.





### 3.4.1 LMA flashes stratified by associated positive and negative CG activity

In this subsection we further stratify the data into flashes associated with negative EUCLID CG strokes (hereby -CG flashes) and flashes associated with positive EUCLID CG strokes (hereby +CG flashes). This section discusses the histograms presented in Fig. 10 to Fig. 13. In those figures the greenish histograms correspond -CG flashes while the reddish histograms correspond to +CG flashes.

There are 713 -CG flashes detected in the dataset and 445 +CG flashes. It should be noted that of the total CG flashes, 73 have both positive and negative CG strokes. Although the existence of bipolar flashes has been documented in the past, at this point we believe that this is mainly due to the limitations in the technique used to stratify the flashes. The necessary tolerance in time and space to associate multiple strokes into flashes may have resulted in the miss-attribution of strokes for flashes very close by in time and/or space. In any case, the proportion of +CG respect to the total number of CG flashes (41%) is significantly higher than that observed on the Säntis tower over a 2-year period (15%) (Romero et al., 2013). That is due to the fact that out of the 8 days analyzed in three of them (10 and 19 July and 1 August) the proportion of +CG flashes is abnormally high (see Table 2). It is in particular noteworthy the percentage of +CG the 19 July (72.6%). In that day large swathes of terrain south of the Säntis tower were affected by hail according to the POH algorithm. In the 1 August there was also extensive hail recorded. The -CG flashes have a total of 178107 VHF sources while +CG flashes have 128446 VHF sources, thus +CG flashes have a more complex structure with an average of 289 sources per flash with respect to the 250 sources associated with -CG flashes.

Regarding the time of occurrence (see Fig. 10 top left panel), for both types of CG flashes there is a well defined peak of occurrence between 16:00 and 18:00. However while the number of -CG and +CG flashes occurring between 16:00 and 17:00 is roughly the same, between 17:00 and 18:00 the number of -CG flashes is double that of +CG flashes. This indicates that individual storms have a preference to produce either one type of flash or the other. The flash duration distribution (Fig. 10 top right panel) is very similar for both -CG and +CG flashes but +CG flashes tend to cover a larger area (Fig. 10 bottom left panel). Indeed the mode and the median of the +CG flashes area coverage is 450 $\mathrm{km}^2$ and 650 $\mathrm{km}^2$ respectively, while that of the -CG flashes is 50 $\mathrm{km}^2$ and 350 $\mathrm{km}^2$. Fig. 10 bottom right panel shows the distribution of number of EUCLID CG strokes per LMA flash, i.e. its multiplicity. +CG flashes mostly have associated a single EUCLID +CG stroke whereas -CG flashes are more likely to have several EUCLID -CG detections associated. The maximum number of EUCLID strokes detected for -CG flashes is 20 while for +CG flashes is just 5.

In terms of source power (not shown), when looking at all VHF sources the distribution is very similar for both +CG flashes and -CG flashes and the median value is the same, 19.5 $\mathrm{dBm}$. However, when looking at the first source the median power of +CG flashes is larger (21.5 $\mathrm{dBm}$ compared to 20.5 $\mathrm{dBm}$).

Regarding the altitude of the VHF sources, there are remarkable differences between the two CG flash types (see Fig. 11 top panels). -CG flashes exhibit a bi-modal distribution with a peak at 7400 $\mathrm{m\,MSL}$ and another one at roughly 4000 $\mathrm{m\,MSL}$. By contrast, +CG flashes have a Gaussian-like distribution with median value 6900 $\mathrm{m\,MSL}$. When looking at the flash origin the contrast is even more marked. +CG flashes have a median altitude of 7500 $\mathrm{m\,MSL}$ while -CG flashes have a larger median altitude of 8000 $\mathrm{m\,MSL}$. Looking at the temperature data from the COSMO model (see Fig. 11 bottom panels) we observe





again a modest decrease in median value when all sources are considered from -CG flashes to +CG flashes (-19°C to -18°C) but a larger decrease when focusing on the flash origin (from -26°C to -22°C). It seems therefore clear +CG flashes tend to have origin at lower levels in the precipitating system.

In terms of the distribution of the polarimetric variables (see Fig. 12), some minor differences can also be noticed. The reflectivity distribution of -CG flashes has a Gaussian-like shape, although skewed towards lower values, with median 37 dBZ while +CG flashes have an almost uniform distribution with median value 34.5 dBZ when all sources are considered. When only first sources are considered, the median is 40.5 dBZ for -CG flashes and 39 dBZ for +CG flashes. The other polarimetric variables have very similar distributions.

The -CG flashes sources can be found in the following regions (histograms not shown): 51.64% in rimed particles, 32.91% in dry hail, 9.62% in dry snow, 0.78% in ice crystals, 2.19% in wet snow, 0.46% in melting hail and 1.94% in rain. 0.51% was in areas without echo classification. When considering only the flash origin, 45.71% were in rimed particles, 44.30% in dry hail, 6.47% in snow, 0.84% in ice crystals, 1.13% in wet snow, 0.14% in melting hail and 0.70% in rain. 0.70% were in areas without hydrometeor classification. The +CG flashes sources can be found in the following areas: 50.95% in rimed particles, 27.68% in dry hail, 13.18% in dry snow, 1.14% in ice crystals, 1.63% in wet snow, 0.52% in melting hail and 2.37% in rain. 0.53% are in areas without echo classification. When considering only the flash origin, 50.11% were in rimed particles, 37.08% in dry hail, 7.64% in snow, 0.67% in ice crystals, 1.35% in wet snow, 0.45% in melting hail and 2.02% in rain. 0.67% are in areas without hydrometeor classification. It is worth highlighting that a higher percentage of +CG flashes originate in the liquid phase region (2.47%) than the corresponding percentage of -CG flashes (0.84%). Also worth noticing is the fact that a larger percentage of -CG flashes have origin in regions where solid hail is the dominant particle (44.30%) with respect to +CG flashes (37.08%). Having said that, as has been the case throughout all the data analysis, the entropy of the hydrometeor classification is rather high (0.4 mode) and most of the radar gates contain 2 or more hydrometeor types in significant proportions.

Fig. 13 examines the most likely combination of hydrometeors at the location of the VHF sources. When looking at all sources, the most likely combinations for -CG flashes are solid hail and rimed particles regardless of the dominant type followed by a combination of rimed particles as dominant and aggregates as second most common. When looking at the flash origin, the most likely combination is clearly dominant solid hail and rimed particles as the second most dominant hydrometeor followed by rimed particles as dominant and solid hail as second most dominant. When looking at +CG flashes the distribution is similar but there are some significant differences. The most likely combination when looking at all VHF sources clearly becomes rimed particles as dominant and solid hail as second most dominant. In second place with similar percentages there is solid hail as dominant and rimed particles as second most dominant and rimed particles as dominant and aggregates as second most dominant. When looking at the flash origin, the most likely combination is rimed particles as dominant and solid hail as second most dominant followed by solid hail as dominant and rimed particles as second most dominant. A significant percentage of flashes have origin in areas where rimed particles are dominant but aggregates are the second most dominant and in areas where the radar volume essentially contains solid hail.



### 3.5 Characteristics of the flashes with origin in the liquid or mixed-phase regions

We focus here our attention on LMA flashes with origin in the liquid or mixed-phase regions. The reason for that is that those flashes are more likely to be of the upward lightning type. For our classification we have considered as belonging to the mixed phase or liquid regions flashes the first VHF source of which was located in areas where the dominant hydrometeor was light

rain, rain, melting hail or wet snow. Flashes located in regions of wet snow are considered to have origin in the mixed phase while flashes with sources in light rain, rain or melting hail are considered to have origin in the liquid region. This section discusses the histograms presented in Fig. 15 to Fig. 19. In those figures, the bluish histogram corresponds to flashes with origin in either the mixed phase or liquid regions, the greenish histograms corresponds to flashes with origin in the mixed phase regions and the reddish histograms correspond to flashes with origin in the liquid phase. In the following we will call

flashes with origin in the liquid phase LP flashes, flashes with origin in the mixed phase MP classes and flashes with origin either in the mixed phase or the liquid phase NSP (non solid phase) flashes.

There are only 241 NSP flashes in the dataset. These flashes generated a total of 31651 VHF sources. Of those, 100 are MP flashes (with 12655 VHF sources) and 141 are LP flashes (with 18996 sources). Regarding their position, Fig. 14 shows that they are mostly distributed in a narrow area going from south-west to north-east in line with the direction of the Alps. It is

interesting to notice that a higher concentration of flashes origins can be seen at the location of the Säntis tower and at another location south of it that we have identified as the Gamsberg area. This backs our assumption that a large percentage of those flashes have origin on the ground.

Regarding the time of occurrence (see Fig. 15 top left panel), it is interesting to notice that those types of flashes are distributed in a more uniform manner than when considering all the flashes. The flash duration distribution (Fig. 15 top right

panel) shows that most flashes are relatively short-lived. The median duration of NSP flashes is 150 ms. They also cover a reduced area with a median of 50 km$^2$ (see 15 bottom panel).

In terms of source power (Fig. 16), when all VHF sources are considered, these flashes have a lower median than the general data, 18.5 dBm. When only the first VHF source in the flash is considered, the median power is 19.5 dBm for MP flashes and 18.5 dBm for LP flashes. It should be noticed that the distribution is not Gaussian-like but has two distinct peaks. A main one

roughly at 20 dBm and a second one around 5 dBm, although this could simply be due to undersampling.

Regarding the altitude of the VHF sources, there are remarkable differences with the global data (see Fig. 17 top panels). As it should be expected NSP flashes are first detected at low altitudes (Median of 3500 m MSL for MP flashes and 3200 m MSL for LP flashes) but their altitude when considering all sources has a bi-modal distribution with a main peak at 3600 m MSL for MP flashes (3800 for LP flashes) and another roughly at 9000 m MSL for both. These data are further confirmed when

looking at the temperature from the model (Fig. 17 bottom panels). When first detected, NSP flashes are located in regions with temperature 0°C or positive but they seem to extend higher up and concentrate in two main layers, one roughly at -5°C and the other at -30°C.

The distribution of the polarimetric variables (see Fig. 18) has significant differences with the global data. The reflectivity has a uniform-like distribution extending from roughly 10 to 50 dBZ, with two barely visible peaks at 10 and 40 dBZ. There



is not enough data to fully characterize the reflectivity at flash origin but it appears to have a Gaussian-like distribution with median 35.5 dBZ for NSP flashes and 32 and 40 dBZ when stratifying into MP and LP flashes respectively. $Z_{dr}$ has a Gaussian-like shape centered at 0 dB when considering all VHF sources. When considering only the flash origin, the distribution is also Gaussian-like but with a positive median of 0.3 dB for NSP flashes (0.3 respectively 0.6 for MP and LP flashes). $\rho_{hv}$ has also a
much wider distribution than the global data, particularly when considering the flash origin. The mode is 0.997 for NSP (0.998 for MP flashes and 0.994 for LP flashes). $K_{dp}$ is skewed towards positive values. The large prevalence of values $2°km^{-1}$ or larger in LP flashes is particularly remarkable.

Fig. 19 top left panel shows the distribution of the dominant hydrometeors at the flash sources location. There are remarkable differences with respect to the global data. When considering NSP flashes, as usual the most common hydrometeor is rimed
particles with 48.38% but it is followed by dry snow (17.58%) and solid hail (10.72%). The proportion of ice crystals is relatively low, only 2.59%. By contrast, a large percentage of sources are located in areas of rain (9.68%) and wet snow (7.95%). When further stratifying the data according to the flash origin, it can be seen that there is a larger proportion of sources located in the mixed phase for MP flashes with respect to LP flashes (12.53% and 4.91% respectively). The most salient feature though is a significant increase of the solid hail proportion of LP flashes (14.30% with respect to the 5.36% of
the MP flashes). What is most remarkable though is that when examining the entropy of the hydrometeor classification (see Fig. 19 middle panels) at the flash origin, it is much higher than that of the global data. Indeed LP flashes have an entropy mode of 0.5 and MP flashes have an even higher entropy mode, 0.6. That translates into a higher proportion of flashes with origin in radar gates containing a mix of 2 and up to 5 hydrometeors (see Fig. 19 lower panels).

Fig. 20 examines the most likely combination of hydrometeors at the location of the VHF sources. Unlike in the other data
analyzed, when all VHF sources are considered, the most common combination is that of rimed particles and snow, followed by rimed particles and dry hail. The most striking feature when looking at the flash origin is that the most dominant combination is a mixture. In the case MP flashes the most likely combination is wet snow and rain, followed by wet snow and rimed particles. LP flashes on the other hand have a mixture of rain and wet snow as the most likely.

## 4   Conclusions

We have presented an analysis of a large dataset of lightning and polarimetric weather radar data collected in the context of a lightning measurement campaign that took place in the summer of 2017 in the area surrounding the Säntis mountain, in northeastern Switzerland. In this campaign, for the first time in the Alps, a lightning mapping array was deployed. This paper focuses on data from 8 days where lightning activity was registered in the immediate vicinity of the Säntis tower. A total of 1586394 VHF sources, corresponding to 12062 flashes (i.e. and average of 132 sources per flash) were detected by the LMA.
In this paper we have investigated the characteristics of the LMA flashes and related their VHF sources to co-located polarimetric radar measurements in order to determine the characteristics of the precipitation systems that enable the initiation and propagation of lightning. We have performed a general assessment and proceeded to stratify the data into the usual categories,




i.e. intra-cloud, cloud-to-ground (positive or negative according to the associated EUCLID strokes) and flashes with origin in the liquid and mixed phase layers as a proxy for upward lightning.

The general data analysis shows that there is a clear diurnal cycle in the days analyzed with most flashes occurring in the late afternoon. Most lightning-producing systems traveled from south-weast to north-east roughly following the foothills of the

Alps. VHF sources are more likely to be detected at altitudes between 3000 and 9000 m MSL. Their origin though, is more likely to be located at altitudes between 7000 to 9000 m MSL. Flashes thus originate in the upper part of convective clouds and either propagate at roughly the same altitude or move towards a lower layer. Most of the flashes originate in areas of high reflectivity (roughly 40 dBZ), low $Z_{dr}$, low $K_{dp}$ and high $\rho_{hv}$. A large majority of flashes originate in regions with a high concentration of particles with a certain degree of riming (either small rimed particles or solid hail) and for the most part they

propagate within such regions. However, these regions are characterized by a high entropy and the radar resolution volume is likely to contain more than one hydrometeor type in significant proportions. The most likely combination of hydrometeors is rimed particles and solid hail, regardless of which one is dominant. Very few flashes originate in the mixed phase or liquid phase of precipitation and the number of VHF sources in those regions is also rather small, suggesting that flashes may transit through them but with a vertical direction.

Most of the examined flashes did not produce an associated EUCLID CG stroke, i.e. they were IC. CG flashes constitute approximately 10% of the total data set. There are significant differences between IC and CG flashes. CG flashes tend to generate more VHF sources, last significantly longer and have a larger projected area. The median altitude at which they are generated is lower than the IC ones and the temperature in the regions of generation is higher, a clear indicator that they are generated lower in the cloud. The co-located polarimetric variables have fairly similar distributions to that of the IC flashes,

except that generally speaking they are wider. The reflectivity at the flash origin location, in particular, has a significantly larger median, suggesting that CG flashes are more likely to occur in regions of higher particle concentration and/or larger particle size. That aspect is further confirmed by an increased proportion of flashes having their origin in areas where solid hail is the dominant hydrometeor. Indeed the most likely combination of hydrometeors at the CG flash origin is one having solid hail as dominant and rimed particles as second most dominant.

We have also attempted to further stratify the LMA flashes associated with EUCLID CG strokes into those generating a positive stroke and those generating a negative stroke. Roughly two thirds of the flashes in the dataset are -CG and one third are +CG. The characteristics of both types of flashes are rather similar although with some significant differences. +CG flashes tend to cover a larger area. -CG flashes are more likely to be associated with multiple EUCLID strokes. The median of the signal power at the origin of the discharge is slightly larger for +CG flashes. The altitude and temperature distributions have

also significant differences. +CG flashes originate at a significantly lower altitude than -CG flashes. When all VHF sources are considered, the distribution of -CG flashes is bi-modal while the distribution of +CG flashes is Gaussian-like. This would suggest that -CG flashes tend to propagate in multiple layers while +CG flashes have a more vertical structure. There are not many differences between the distribution of polarimetric variables of -CG and +CG flashes. The most remarkable difference is in the distribution of the reflectivity when all VHF sources are considered. -CG flashes have a Gaussian-like shape while +CG

flashes have an almost uniform distribution. Again, that would suggest that -CG flashes are more likely to propagate in the





cloud layer where they have originated whereas +CG flashes have a more vertical structure. As with all the data analyzed, the entropy of the hydrometeor classification is rather high and it is very likely to have multiple hydrometeor types with significant proportions within a radar resolution volume. The proportion of dominant hydrometeors is similar for both types of flashes. The main differences are a larger percentage of -CG flashes having origin in areas of solid hail. However, in both cases the

most likely combination of hydrometeors at the flash origin is solid hail and rimed particles, albeit in different proportions.

    We have also examined the characteristics of LMA flashes having their origin in the liquid and mixed phase layers. The rationale for that is that there is an increased likelihood that those flashes are upward lightning. Only 241 flashes in the dataset had origin in the mixed phase or liquid layers and of those, 141 had origin in the liquid layer. One aspect that sets apart this type of flashes is that their origins are not randomly distributed within the path of the storm but we have identified areas of higher

concentration. One of them is the Säntis tower and another is the Gamsberg mountain. That is a strong indicator that indeed these flashes originate by interactions with the terrain and man-made objects. These flashes tend to be short-lived and occupy a reduced projected area, which would indicate a vertical orientation. As it would be expected, they are first detected at low altitudes but their altitude distribution is bi-modal with a secondary peak at roughly 9000 m MSL, a rather high altitude within the precipitating system. The distribution of the polarimetric variables is also fairly different from the rest of the data analyzed.

Overall that distributions suggest that this sort of flashes encounter a larger variety of precipitation regimes along their path. This is further confirmed by the fact that the entropy of the hydrometeor classification is even higher than in the previously examined cases, particularly at flash origin, and it is rather common to have more than 2 species in significant proportions at flash origin. Unlike with the rest of the data examined, rimed particles or solid hail are not part of the most common species in the mixture at the location of the flash origin, the most common being a mixture of wet snow and rain. However, when

examining all the VHF sources, the most common mixture at the VHF source position is rimed particles and snow. Solid hail, on the other hand, does not have such a significant presence as with the global data. This would indicate that this sort of flashes can be initiated even in moderate convection conditions.

    From our analysis we can conclude that a systematic detection of lightning with polarimetric radar data is not possible but radar data can be extremely useful to indicate regions with favorable conditions for lightning initiation. A high concentration,

i.e. high reflectivity, of rimed particles and/or solid hail and a high entropy of the hydrometeor classification are a primary indicator of lightning activity. Whether the generated flashes are IC or CG it is not something that can be readily determined by just observing the polarimetric radar data although there are indicators that suggests that CG flashes tend to generate at a lower cloud level and, therefore, looking at the vertical structure of the precipitating system can provide hints on whether the flashes will more likely be contained within the cloud or propagate to the ground. Upward lightning poses another challenge

because this type of flashes have not shown such clear radar signatures as the rest of the examined categories and seem to be driven mostly by the interaction of the storm with the terrain and man-made structures. Nevertheless, also in that case a degree of convection seems to be necessary to initiate a flash. In any case, to estimate the probability of this sort of flashes it seems clear that either the orography (including man-made structures) or a climatology has to be included in any flash nowcasting system.




*Code and data availability.* Code used to post-process the radar data is available on github https://github.com/meteoswiss-mdr. Data is available on request by contacting the authors.

*Author contributions.* JFV performed the radar data processing and the data analysis contained in this paper. NB and JG contributed to the radar data processing and data interpretation. OvV, DR, JM, NP, AS, AM, MA, MR and FR deployed the LMA network and processed its data. UG and AH advised on the content of the manuscript. JFV, with contributions from all authors, prepared the manuscript.

*Competing interests.* The authors declare that they have no conflict of interest.

*Acknowledgements.* This work was partially supported by the Swiss National Science Foundation (Project No. 200020_175594) and the European Union's Horizon 2020 research and innovation program under grant agreement No 737033-LLR.



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



**Table 1.** Location of LMA sensors

| Station | Latitude [°] | Longitude [°] | Altitude [m MSL] |
|---------|--------------|---------------|------------------|
| Kronberg | 47.2917 | 9.3276 | 1655 |
| Gonten | 47.3170 | 9.3504 | 903 |
| STBO | 47.1713 | 9.2543 | 1340 |
| Urnäsch | 47.3193 | 9.2948 | 835 |
| Schwagalp | 47.2539 | 9.3201 | 1288 |
| Säntis | 47.2493 | 9.3276 | 2499 |





**Table 2.** Analyzed days with some of their general characteristics. The LMA domain refers to the yellow area in Fig. 1. The cells considered have a life span of at least 3 radar time steps (i.e. 15 min) and have been present within the domain for at least 3 time steps

| Days examined | LMA | LMA Flashes | LMA Sources | Säntis | CG | -CG | +CG | bipolar CG? | %+CG |
|---|---|---|---|---|---|---|---|---|---|
| 2017.06.29 | 6 | 250 | 41721 | 13 | 44 | 42 | 2 | 0 | 4.5 |
| 2017.06.30 | 5 | 1155 | 285291 | 0 | 30 | 25 | 5 | 0 | 16.7 |
| 2017.07.10 | 5 | 1339 | 171743 | 3 | 32 | 27 | 11 | 6 | 34.4 |
| 2017.07.14 | 5 | 263 | 20897 | 1 | 10 | 8 | 2 | 0 | 20.0 |
| 2017.07.18 | 5 | 635 | 115617 | 3 | 24 | 19 | 6 | 1 | 25.0 |
| 2017.07.19 | 5 | 2250 | 202343 | 7 | 164 | 54 | 119 | 9 | 72.6 |
| 2017.07.22 | 4 | NA | NA | 0 | NA | NA | NA | NA | NA |
| 2017.07.24 | 3 | NA | NA | 0 | NA | NA | NA | NA | NA |
| 2017.07.25 | 3 | NA | NA | 0 | NA | NA | NA | NA | NA |
| 2017.07.30 | 6 | 960 | 138445 | 0 | 16 | 14 | 2 | 0 | 12.5 |
| 2017.08.01 | 6 | 5210 | 610337 | 2 | 765 | 524 | 298 | 57 | 40.0 |
| 2017.08.05 | 5 | NA | NA | 1 | NA | NA | NA | NA | NA |
| 2017.08.08 | 5 | NA | NA | 1 | NA | NA | NA | NA | NA |
| 2017.08.09 | 5 | NA | NA | 1 | NA | NA | NA | NA | NA |
| 2017.08.15 | 5 | NA | NA | 1 | NA | NA | NA | NA | NA |





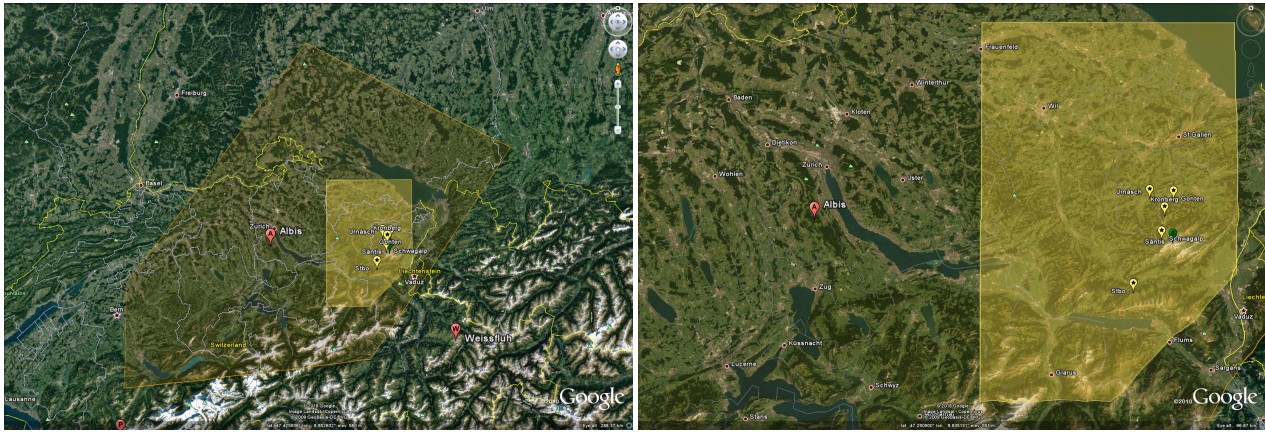

**Figure 1.** Left: Approximate extend of the maximum area covered by the LMA (Orange poligon). The yellow area shows the region with more comprehensive coverage. Radar positions are marked by red dots while the position of the LMA sensors is marked by yellow dots. The Säntis tower is marked by a green dot. Right: Zoom over the best covered area.

| Event | Jun | | July | | | | | | | | | | | | | | | | | | | | | | | | | | | | | | | August | | | | | | | | | | | | | | |
|---|---|---|---|---|---|---|---|---|---|---|---|---|---|---|---|---|---|---|---|---|---|---|---|---|---|---|---|---|---|---|---|---|---|---|---|---|---|---|---|---|---|---|---|---|---|---|
| | 29 | 30 | 1 | 2 | 3 | 4 | 5 | 6 | 7 | 8 | 9 | 10 | 11 | 12 | 13 | 14 | 15 | 16 | 17 | 18 | 19 | 20 | 21 | 22 | 23 | 24 | 25 | 26 | 27 | 28 | 29 | 30 | 31 | 1 | 2 | 3 | 4 | 5 | 6 | 7 | 8 | 9 | 10 | 11 | 12 | 13 | 14 | 15 |
| Lightning | | | | | | | | | | | | | | | | | | | | | | | | | | | | | | | | | | | | | | | | | | | | | | | | |
| Lightning 2 Km from tower | | | | | | | | | | | | | | | | | | | | | | | | | | | | | | | | | | | | | | | | | | | | | | | | |
| Hail (90% prob) | | | | | | | | | | | | | | | | | | | | | | | | | | | | | | | | | | | | | | | | | | | | | | | | |
| Hail 20 km from tower | | | | | | | | | | | | | | | | | | | | | | | | | | | | | | | | | | | | | | | | | | | | | | | | |

**Figure 2.** Overview of relevant events within the LMA reduced domain occurred during the 2017 campaign. Hail probability is derived from the POH radar product. Occurrence of relevant events is marked in green.



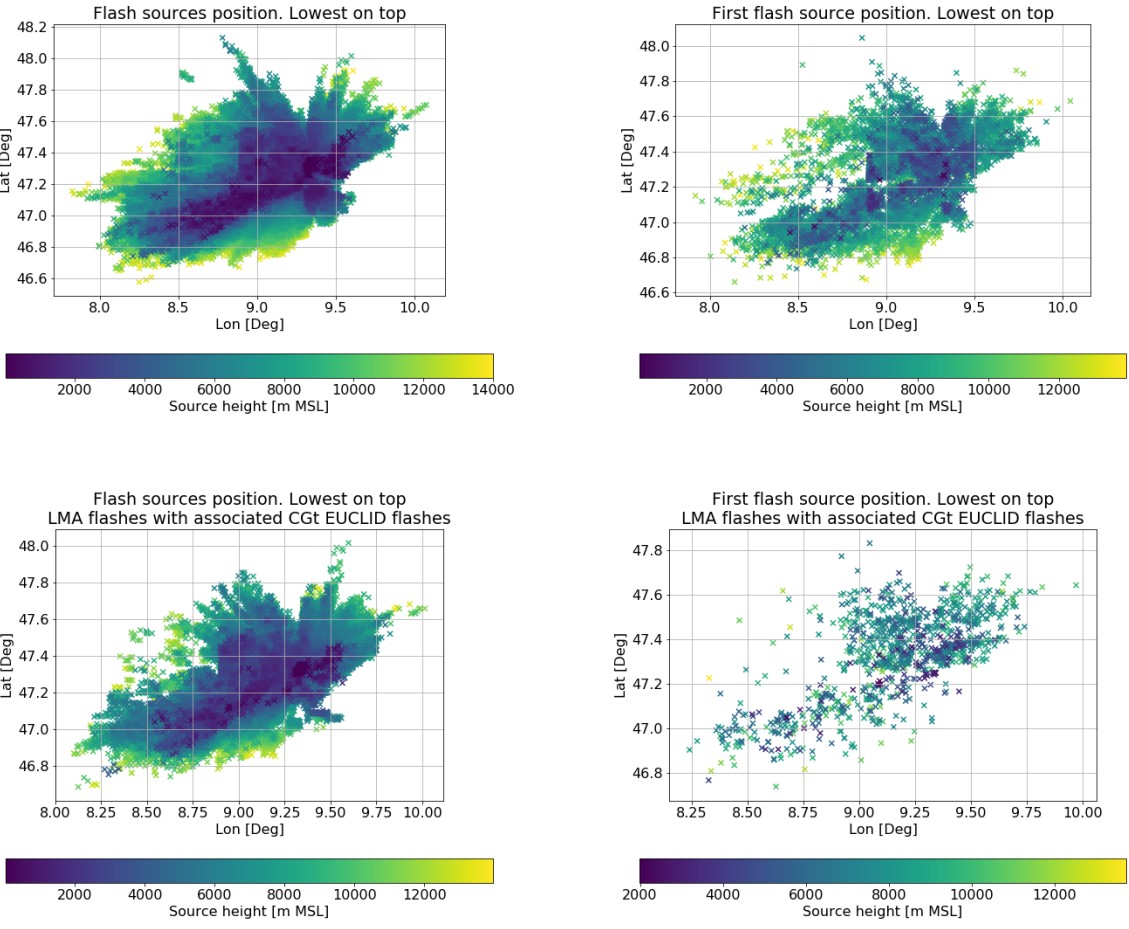

**Figure 3.** Position of detected VHF sources for the days examined. Left: All VHF sources, Right: Only first sources of each flash. Top : All data, Bottom: flashes with associated EUCLID CG strokes



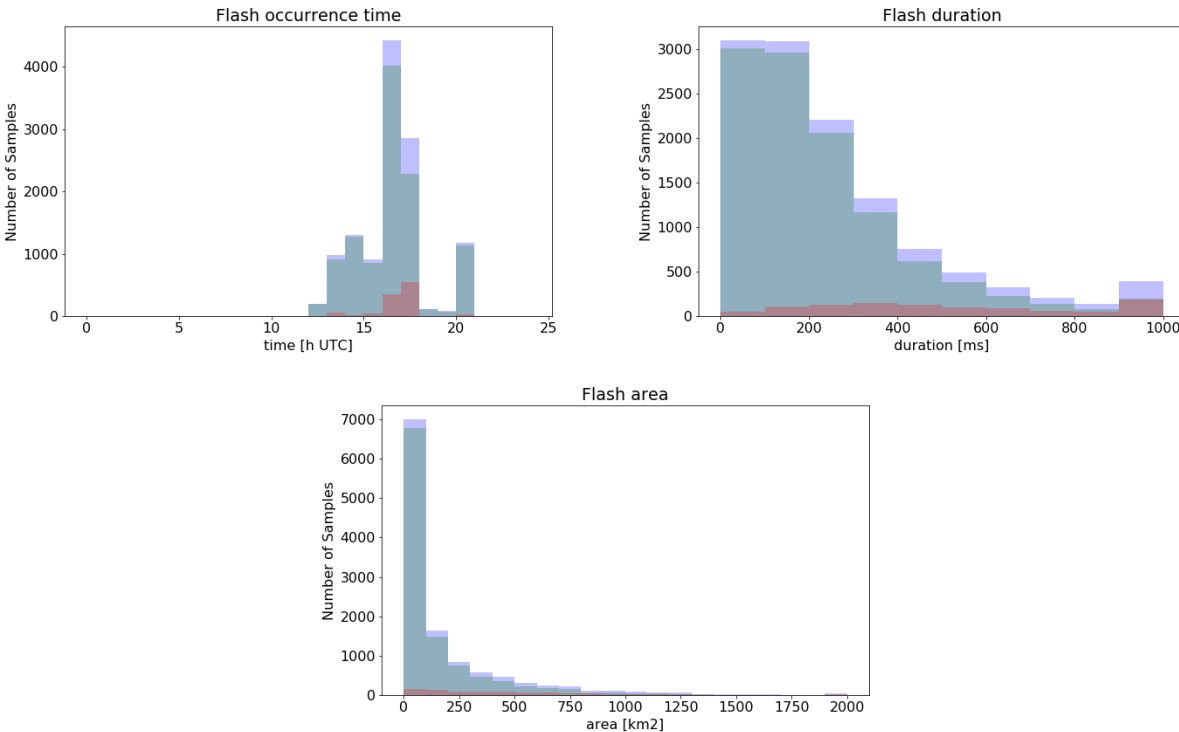

**Figure 4.** Distribution of the LMA flashes for all days analyzed: Top left: time of occurrence, Top right: duration, Bottom: 2D projection area over the day for all days analyzed. Bluish: All data, Greenish: Flashes without associated EUCLID CG strokes, Reddish: Flashes with associated EUCLID CG strokes.





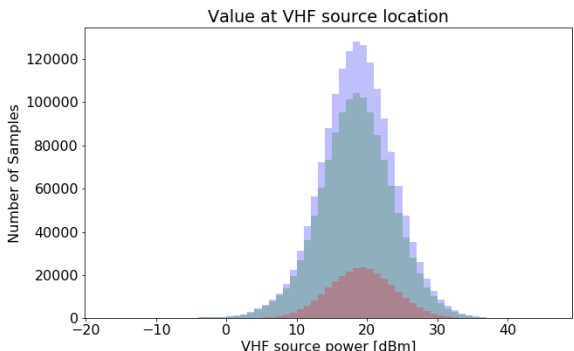
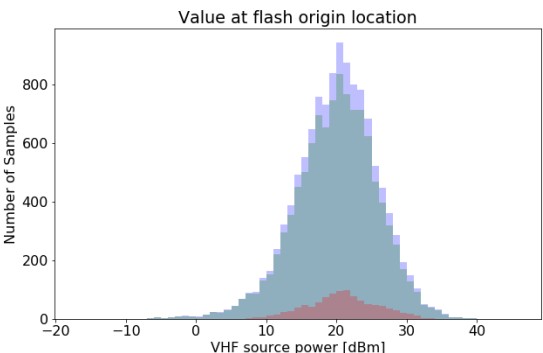

**Figure 5.** Histogram of VHF sources power for all days analyzed. Left: All sources. Right: Only first sources of each flash. Bluish: All data, Greenish: Flashes without associated EUCLID CG strokes, Reddish: Flashes with associated EUCLID CG strokes.




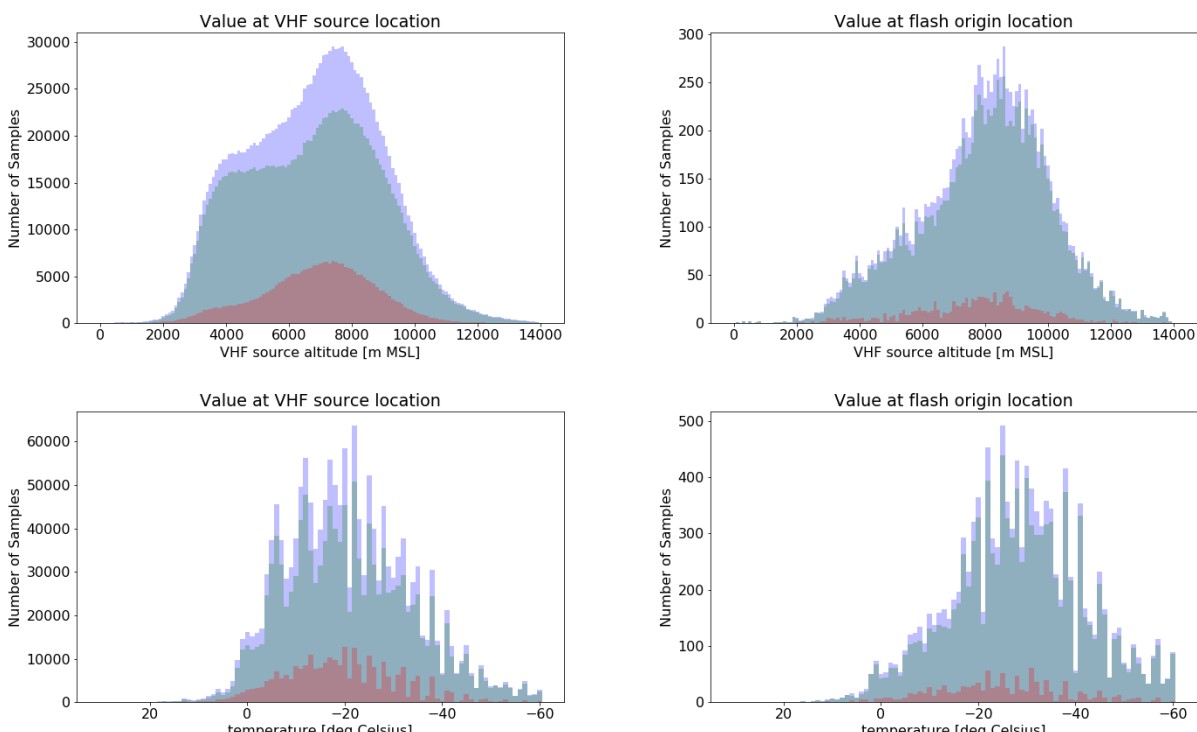

**Figure 6.** Histogram during all days analyzed of: Top left: VHF sources altitude. Top right: Only first sources of each flash altitude. Bottom left: All sources model air temperature. Bottom right: First sources model air temperature. Bluish: All data, Greenish: Flashes without associated EUCLID CG strokes, Reddish: Flashes with associated EUCLID CG strokes.




**Figure 7.** Histogram during all days analyzed of from top to bottom: $Z_h$, $Z_{dr}$, $\rho_{hv}$, $K_{dp}$. Left: All sources. Right: First source only. Bluish: All data, Greenish: Flashes without associated EUCLID CG strokes, Reddish: Flashes with associated EUCLID CG strokes.





**Figure 8.** Histogram during all days analyzed of from top to bottom: Dominant hydrometeor at the radar gate colocated with the VHF source position (The numbers correspond to the following: 0=NC, 1=DS, 2=CR, 3=LR, 4=RP, 5=RN, 6=VI, 7=WS, 8=MH, 9=IH), hydrometeor classification derived entropy at the radar gate colocated with the VHF source position, Number of hydrometeors types with significant presence at the radar gate colocated with the VHF source position. Left: All sources. Right: First source only. Bluish: All data, Greenish: Flashes without associated EUCLID CG strokes, Reddish: Flashes with associated EUCLID CG strokes.





**Figure 9.** 2D-histogram of the type of the dominant and second most dominant hydrometeor at the radar gate colocated with the VHF source position (The numbers correspond to the following: 0=NC, 1=DS, 2=CR, 3=LR, 4=RP, 5=RN, 6=VI, 7=WS, 8=MH, 9=IH). From top to bottom: All data, flashes without associated EUCLID CG strokes, flashes with associated EUCLID CG strokes. Left: All VHF sources in flash, right: only first VHF sources.



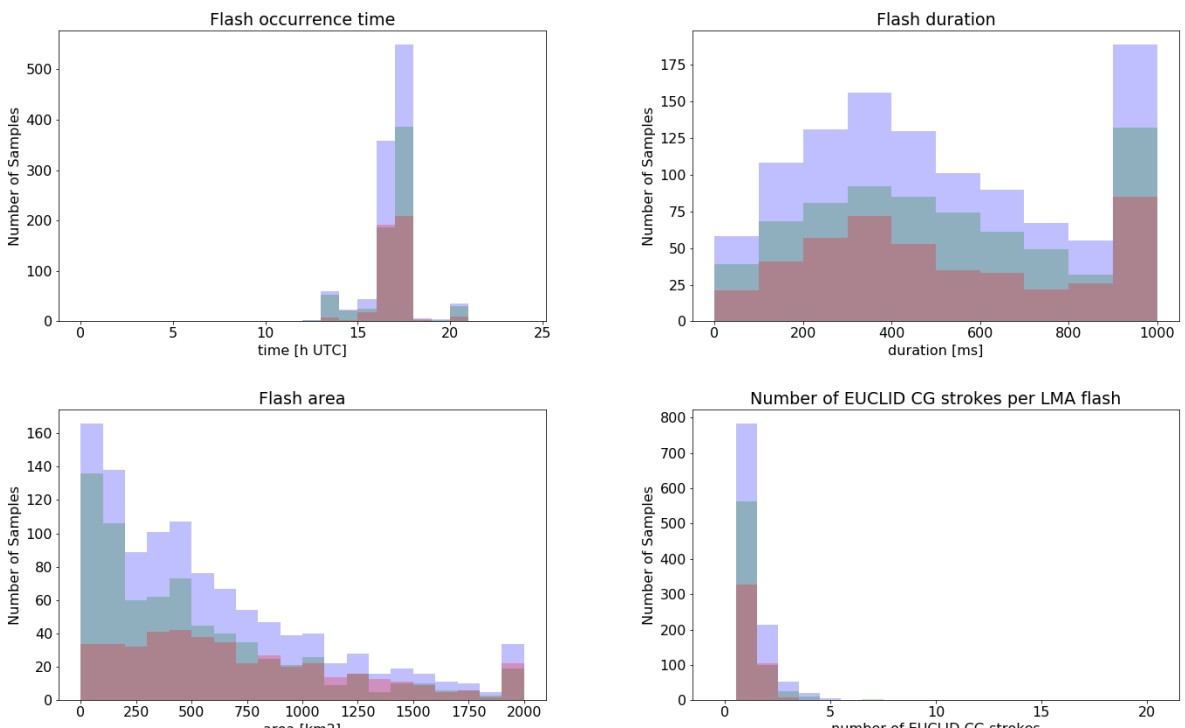

**Figure 10.** Distribution of the flashes for all days analyzed: Top left: time of occurrence, Top right: duration, Bottom left: 2D projection area over the day for all days analyzed. Bottom right: Number of detected EUCLID CG strokes per LMA flash. Bluish: All flashes with associated EUCLID CG strokes, Greenish: Flashes with associated EUCLID -CG strokes, Reddish: Flashes with associated EUCLID +CG strokes.





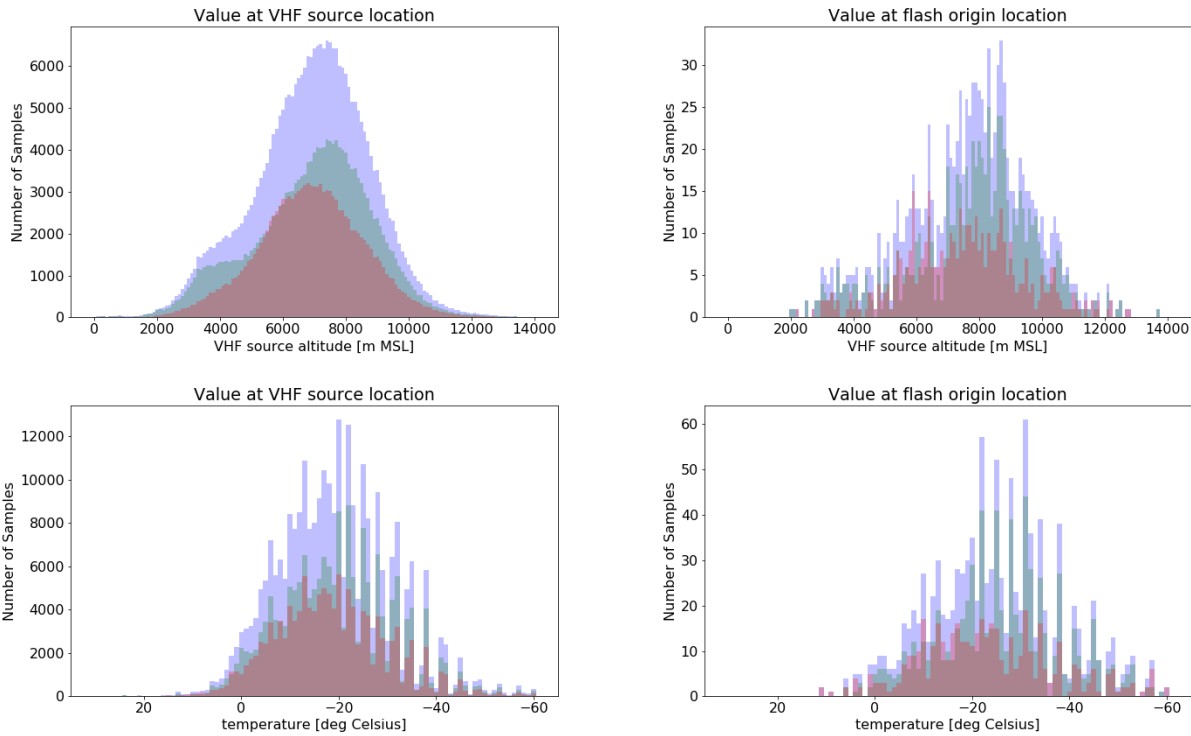

**Figure 11.** Histogram during all days analyzed of: Top left: VHF sources altitude. Top right: Only first sources of each flash altitude. Bottom left: All sources model air temperature. Bottom right: First sources model air temperature. Bluish: All flashes with associated EUCLID CG strokes, Greenish: Flashes with associated EUCLID -CG strokes, Reddish: Flashes with associated EUCLID +CG strokes.





**Figure 12.** Histogram during all days analyzed of from top to bottom: $Z_h$, $Z_{dr}$, $\rho_{hv}$, $K_{dp}$. Left: All sources. Right: First source only. Bluish: All flashes with associated EUCLID CG strokes, Greenish: Flashes with associated EUCLID -CG strokes, Reddish: Flashes with associated EUCLID +CG strokes.





**Figure 13.** 2D-histogram of the type of the dominant and second most dominant hydrometeor at the radar gate colocated with the VHF source position (The numbers correspond to the following: 0=NC, 1=DS, 2=CR, 3=LR, 4=RP, 5=RN, 6=VI, 7=WS, 8=MH, 9=IH). From top to bottom: all flashes with associated EUCLID CG data, flashes with associated EUCLID -CG strokes, flashes with associated EUCLID +CG strokes. Left: All VHF sources in flash, right: only first VHF sources.







**Figure 14.** Position of detected VHF sources for the days examined. Left: All VHF sources, Right: Only first sources of each flash. From top to bottom: Flashes with origin in the liquid and mixed phase regions, flashes with origin in the mixed phase region, fFlashes with origin in the liquid region





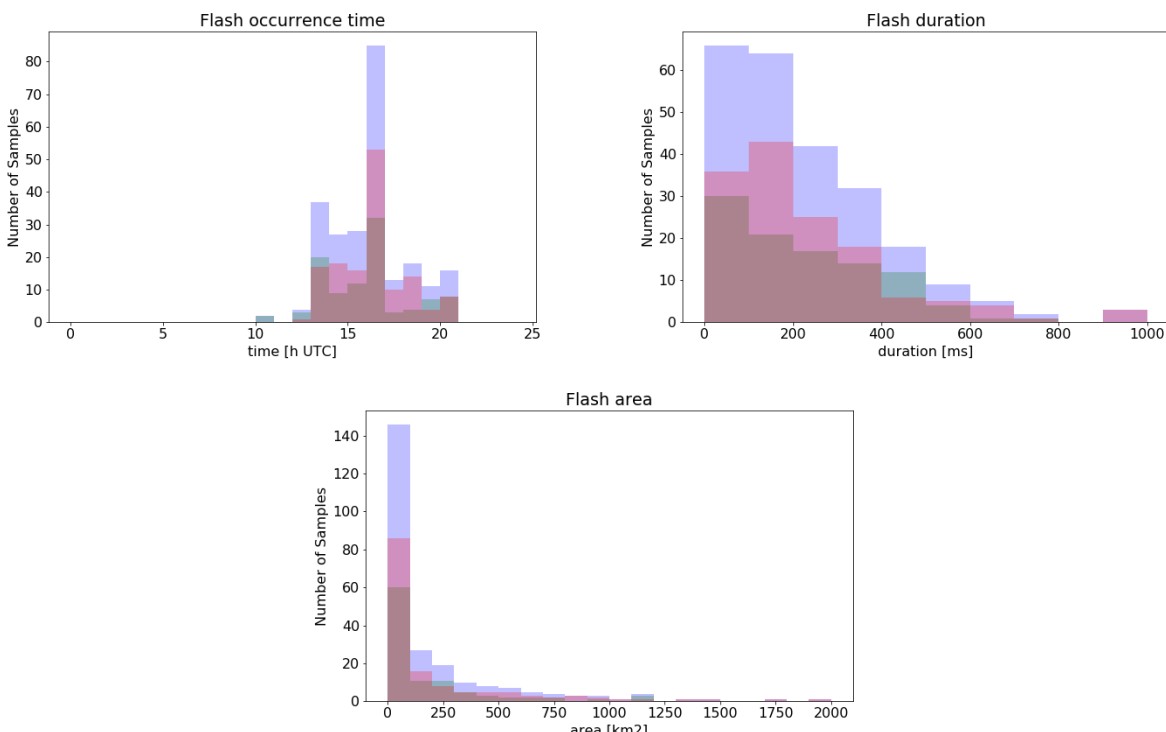

**Figure 15.** Distribution of the flashes for all days analyzed: Top left: time of occurrence, Top right: duration, Bottom: 2D projection area over the day for all days analyzed. Bluish: Flashes with origin in the liquid and mixed phase regions, Greenish: Flashes with origin in the mixed phase region, Reddish: Flashes with origin in the liquid region.




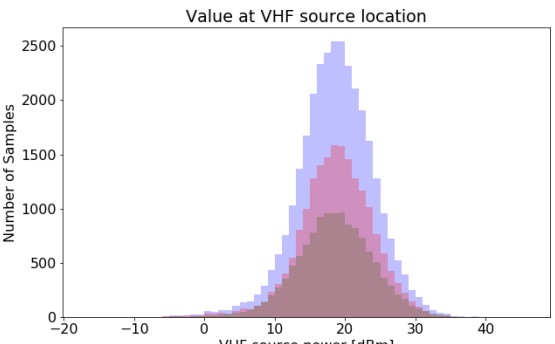
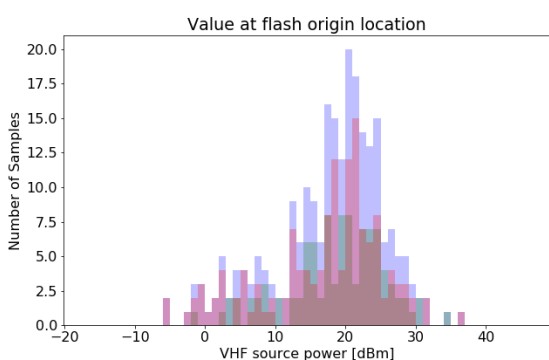

**Figure 16.** Histogram of VHF sources power for all days analyzed. Left: All sources. Right: Only first sources of each flash. Bluish: Flashes with origin in the liquid and mixed phase regions, Greenish: Flashes with origin in the mixed phase region, Reddish: Flashes with origin in the liquid region.





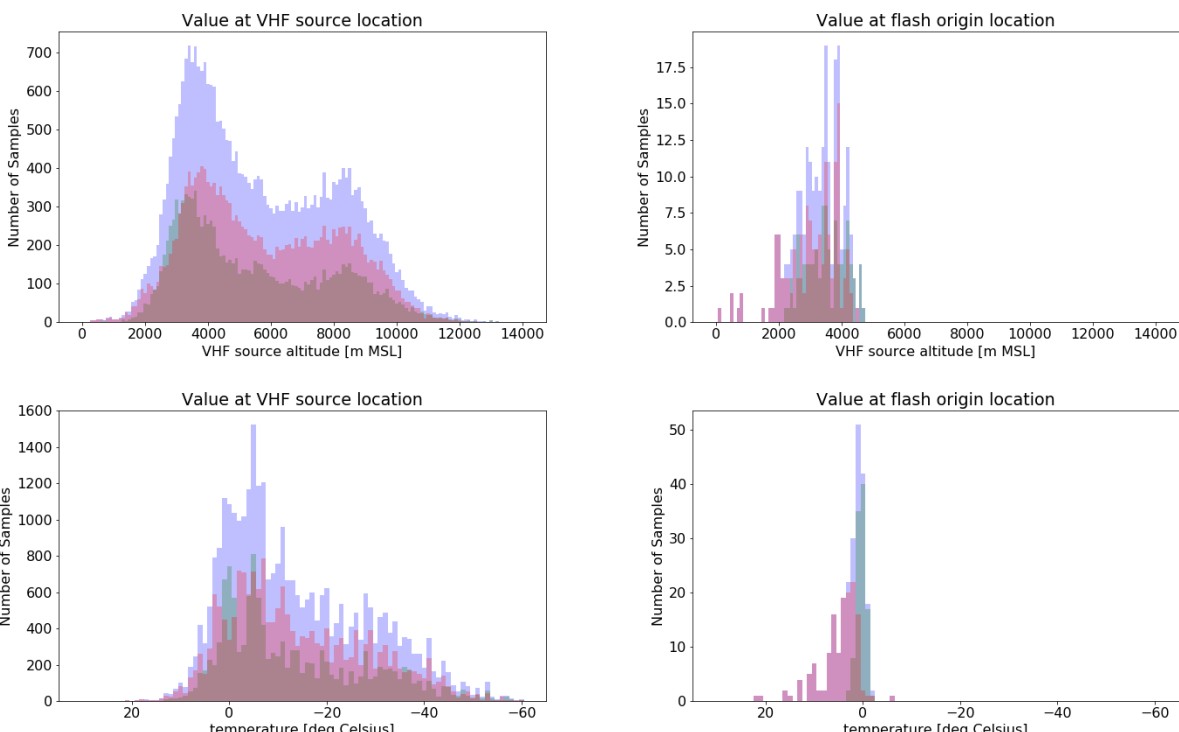

**Figure 17.** Histogram during all days analyzed of: Top left: VHF sources altitude. Top right: Only first sources of each flash altitude. Bottom left: All sources model air temperature. Bottom right: First sources model air temperature. Bluish: Flashes with origin in the liquid and mixed phase regions, Greenish: Flashes with origin in the mixed phase region, Reddish: Flashes with origin in the liquid region.







**Figure 18.** Histogram during all days analyzed of from top to bottom: $Z_h$, $Z_{dr}$, $\rho_{hv}$, $K_{dp}$. Left: All sources. Right: First source only. Bluish: Flashes with origin in the liquid and mixed phase regions, Greenish: Flashes with origin in the mixed phase region, Reddish: Flashes with origin in the liquid region.





**Figure 19.** Histogram during all days analyzed of from top to bottom: Dominant hydrometeor at the radar gate colocated with the VHF source position (The numbers correspond to the following: 0=NC, 1=DS, 2=CR, 3=LR, 4=RP, 5=RN, 6=VI, 7=WS, 8=MH, 9=IH), hydrometeor classification derived entropy at the radar gate colocated with the VHF source position, Number of hydrometeors types with significant presence at the radar gate colocated with the VHF source position. Left: All sources. Right: First source only. Bluish: Flashes with origin in the liquid and mixed phase regions, Greenish: Flashes with origin in the mixed phase region, Reddish: Flashes with origin in the liquid region.





**Figure 20.** 2D-histogram of the type of the dominant and second most dominant hydrometeor at the radar gate colocated with the VHF source position (The numbers correspond to the following: 0=NC, 1=DS, 2=CR, 3=LR, 4=RP, 5=RN, 6=VI, 7=WS, 8=MH, 9=IH). From top to bottom: Flashes with origin in the liquid and mixed phase regions, flashes with origin in the mixed phase region, flashes with origin in the liquid region. Left: All VHF sources in flash, right: only first VHF sources.