# Peer review of "Polarimetric radar characteristics of lightning initiation and propagating channels"

_Atmospheric Measurement Techniques, 2019_

## Referee Comment (RC1) · Anonymous Referee #1 · 26 Feb 2019

GENERAL COMMENT This paper presents a detailed study on lightning strokes in Switzerland, in relation with hydrometeor classification from dual-polarization weather radar. The topic is original and certainly interesting for a wide range of cloud physics applications. The paper is in general well written, with a comprehensive introduction including the scientific background and a proper description of the instrumentation and analysis methods adopted. The data analysis is highly descriptive, including a consistent amount of results organized according to partitions including IC, CG, positive and negative strokes, and finally lightnings originating at lower levels (mixed-phase region). The illustrated results are very interesting and worth to be published on AMT journal. As also detailed in the specific comments below, I found this part (section 3)

[Figure]

not enough fluent due to an overwhelming amount of numbers in the text. For this reason, I suggest making use of tables to organize at least some of these numbers and try to improve the fluency focusing on the relevant qualitative behaviors. In addition, while the introduction is properly developed with many relevant references, in the data analysis I missed references helping to understand whether the results are supported by previous findings or the authors are showing some phenomenological behavior not yet comprehensively studied, lacking physical explanation. This is my only major concern, which I'm sure could be easily addressed by the authors. Providing a better context to these results would help the paper readability, and provide other researchers a more solid background to work with lightning data in association with dual-polarization measurements.

Consider adding a figure showing an example of radar hydrometeor classification. A vertical cut, possibly with lightnings over plotted, would greatly help readers not familiar with radar classification, and may serve as an effective visual introduction to the main topic of the paper.

SPECIFIC COMMENTS AND MINOR CORRECTIONS

- When discussing the hydrometeor type in relation with the lightnings, percentages are given to represent the relative occurrence of given particle types. The first comment I have is that providing these numbers with two decimals accuracy is misleading. We know that hydrometeor classification is subject to many assumptions and uncertainties, so I would recommend providing these percentages with a lower level of accuracy (rounding to integers may be enough in general). The second comment is that there are large portions of the manuscript with a detailed listing of these percentages (e.g. page. 11, lines 3-9) which makes reading a bit hard. I recommend moving as much as possible those numbers to tables and leave in the text only the most relevant ones, focusing on the relevant qualitative behavior.

- P.6, Line 24: "ZPhi" method should read "ZPHI", and the proper reference should be
the Testud paper:

Testud, J., E. Le Bouar, E. Obligis, and M. Ali-Mehenni, 2000: The rain profiling algorithm applied to polarimetric weather radar. J. Atmos. Oceanic Technol., 17, 332–356.

Indeed, Ryzhkov used the ZPHI method for rainfall estimation using the same ZPHI method as in Testud et al., 2000 for deriving specific attenuation.

- P. 7, Lines 9-10: according to Besic et al., 2018, if there is a dominant hydrometeor type, the entropy is low (close to 0), while in case of mixtures, the entropy is higher (up to 1). Please explain and/or correct why it is here stated the opposite ("..within the radar resolution volume there is a clearly dominant hydrometeor type (entropy 1) or if it is an heterogeneous mixture without any dominant hydrometeor type (entropy 0)."). Also check that the values of entropy discussed through the paper are consistent with the correct definition.

- P.7, Line 29. "extend" -> "extent"

- P. 9, L25: "The reflectivity data (topmost panels in Fig. 7) of all sources show a bi-modal distribution" Any idea why? Could there be a relation with the bimodality of lightning data?

- Please check the label for aggregates, sometimes it is referred to as "AG" (P. 6, L. 32), but in fig. captions it is often called "DS".

- P. 13, Lines 11-14. Here the authors hint at a possible relation between the very high proportion of positive lightnings and hail at the ground. Please add proper references.

- P. 15, Lines 3-4: "For our classification we have considered as belonging to the mixed phase or liquid regions flashes the first VHF source of which was located in areas where the dominant hydrometeor..." I suggest rewording, i.e. "For our classification we have considered as belonging to the mixed phase or liquid regions flashes whose first VHF source was in areas where the dominant hydrometeor.."

- P. 16, Lines 6-7: have you analysed in detail the "origin" of these 2 deg/km Kdp values (lower-right panel in fig. 18)? Such anomaly in the distribution is quite suspect, could it be due to some artifact in the data processing?

- P.17, Line 2: "flashes with origin in the liquid and mixed phase layers as a proxy for upward lightning". Is there a proper reference to add here? Or is it just a "common sense" expectation?

- P. 17, Lines 20-22: I also suggest considering "riming" (which implies the presence of supercooled water) in addition to particle concentration/size to explain the higher reflectivity.

- Fig. 1. Either add a length scale in the image, or at least mention the size of the domain in the caption. A map with lat, lon axes would be more useful for easier interpretation of fig. 3.

- Fig. 2 Need larger font for the dates.

- Fig. 4 (and all other similar histograms). Please add a legend with the color meaning (in addition to mentioning in the caption), it is sufficient in the first panel.

- Fig. 8, top panels: it may be better to add the hydrometeor type (e.g. "RN",.. ) directly on the x-axis (so the reader is not forced to keep switching between the number and the label), or at least mark the most relevant hydrometeor types directly on the histogram bars.

- Fig. 9: same as above, please mark the hydrometeor type on the axes, try to avoid using the numeric index. Suggestion: consider using a monochromatic color scale (e.g. gray scale), with light tone for low occurrence and dark tone for higher occurrence. I argue this may improve the readability (this is just a suggestion!). The current dark blue really dominates the visual impact.

---

## Referee Comment (RC2) · Anonymous Referee #2 · 27 Feb 2019

The paper presents comprehensive statistical analysis of lightning data associated with concurrent polarimetric radar observations in the Swiss Alps. There is no doubt that the manuscript contains rich information about statistical characteristics if intra-cloud and cloud-to-ground flashes such as their intensity, duration, area, altitude and temperature intervals where flashes originated, etc. It comes at no surprise that both IC and CG flashes mostly originate in the areas of dry graupel (called rimed particles in the manuscript) and hail well above the freezing level in sufficiently deep convective clouds. The authors relate lightning flashes to the output of the MeteoSwiss semi-supervised polarimetric classification algorithm and even estimate the entropy of hydrometeor classification in the flash locations. I am not sure that the use of such a "big gun" as the
polarimetric classifier is fully justified in this context. Indeed, polarimetric radar variables such as ZDR, KDP, and hv bear very little classification potential in cold parts of convective storms unless large hail growing in a wet growth regime is observed in the cloud. In fact, discrimination between snow, graupel, and hail aloft is almost exclusively made based on the radar reflectivity factor. Radar reflectivity of hail is larger than the one of graupel and snow. Large entropy simply means high variability of Z in a given spatial domain that spans typical range intervals of snow, graupel, and hail. I am surprised by the fact that the histograms of ZDR in the regions of lightning initiation above the freezing level are almost perfectly symmetric around 0 dB value. Graupel and hail – major source of lightning flashes – can grow only in sufficiently strong convective updrafts commonly manifested by the ZDR columns. Various researchers report close association of lightning locations and ZDR columns which is not examined and even mentioned in the paper. At the same time, there is apparent sign of nozero positive KDP indicated in the histograms in Figs. 7 and 12. What is the origin of these positive values of KDP in cold parts of convective clouds? Horizontally oriented ice crystals in the proximity of graupel and hail or the tops of KDP columns? In the latter situation, ZDR columns with noticeably positive ZDR should be also observed.

---

## Referee Comment (RC3) · Anonymous Referee #3 · 5 Mar 2019

The present study aims at combining LMA and polarimetric radar observations to infer relationships between lightning activity and cloud microphysics. It is based on a significant dataset collected in the northeastern part of Switzerland over a period of two and a half months during summer 2017. Overall this study provides interesting findings regarding potential interactions between the microphysical cloud structure and lightning initiation/propagation. It is, however, extremely descriptive and definitely lacks some physical interpretation to support the results. Section 3, for instance, mostly consists in a discussion of endless series of numbers and percentages / histograms with no interpretation and no comparison against previous findings. Authors also lack perspective on the performance of their classification algorithm, especially as we all know that

dual-polarimetric HCA are subject to large uncertainties (low percentage values such as discussed in the paper don't really make sense and should definitely be avoided). Actually I believe that Section 3 should be simplified and rewritten to highlight the most significant results as it is very difficult to comment on the results as they stand.

Specific comments:

More details are needed regarding the Hydrometeor Classification Algorithm used in this study. Authors should also discuss the uncertainty associated with their method, especially regarding hail/graupel identification. Actually I am quite surprised by the high proportion of lightning initiation and propagation that takes place in hail. Most past studies have found that graupel is by far (70-90%) the preferred environment for lighting initiation. Also, author are using a C-band radar. However we know that at such frequency hail identification might be a problem due to resonance scattering effects. Please comment.

Radar scientists claim to be able to identify as many hydrometeor types as possible. However, due to the high level of uncertainty in HID, it would be more realistic to use less categories. This would also ease the interpretation of the results.

As mentionned previously, this paper lacks physical interpretation of the results. Results should be compared with the literature (especially regarding hail) and more details should be given about the role of the microphysical environment in the initiation/propagation of lightning. Also what is the influence of orography on lightning? As mentionned by the authors, the particularity of this study is that it takes place over very complex terrain. Hence the effect of mountains should be considered when analyzing the results.

The spatial extension of the domain of analysis seems rather small to me. LMA data can potentially be used up to 100/150 km without any problem. Please comment.

I do not see the need for 20 figures. Most of the results shown in Figs. 4 to 20

could be summarized in a few tables. Instead, authors should include the analysis of two contrasting events in their study and discuss the storm structures, associated horizontal/vertical cross-sections of reflectivity / polarimetric moments, and location of LMA sources within the storms.

A detailed comparison between EUCLID and LMA detection capability would be interesting.

A discrimination between convective and statiform regions of the storms would also help interpreting the results.

---

## Author Comment (AC1) · 18 Apr 2019

The response to the reviewer and the modified manuscript are provided as supplement material.

Please also note the supplement to this comment:
https://www.atmos-meas-tech-discuss.net/amt-2019-31/amt-2019-31-AC1-supplement.zip

———————————————

---

## Author Response (AR1)

**Response to reviewer 1 of amt-2019-31**

In this document we provide answers to the comments of reviewer 1 of the paper amt-2019-31. Our answers to the reviewer are given in *italic* font. Proposed changes to the manuscript are highlighted in blue color.

GENERAL COMMENT

This paper presents a detailed study on lightning strokes in Switzerland, in relation with hydrometeor classification from dual-polarization weather radar. The topic is original and certainly interesting for a wide range of cloud physics applications. The paper is in general well written, with a comprehensive introduction including the scientific background and a proper description of the instrumentation and analysis methods adopted. The data analysis is highly descriptive, including a consistent amount of results organized according to partitions including IC, CG, positive and negative strokes, and finally lightning originating at lower levels (mixed-phase region). The illustrated results are very interesting and worth to be published on AMT journal.

*We thank the reviewer for his/her kind words and thorough review*

As also detailed in the specific comments below, I found this part (section 3) not enough fluent due to an overwhelming amount of numbers in the text. For this reason, I suggest making use of tables to organize at least some of these numbers and try to improve the fluency focusing on the relevant qualitative behaviors. In addition, while the introduction is properly developed with many relevant references, in the data analysis I missed references helping to understand whether the results are supported by previous findings or the authors are showing some phenomenological behavior not yet comprehensively studied, lacking physical explanation. This is my only major concern, which I'm sure could be easily addressed by the authors. Providing a better context to these results would help the paper readability, and provide other researchers a more solid background to work with lightning data in association with dual-polarization measurements.

*We have added a subsection Discussion in section 3 to put our results into context*

Consider adding a figure showing an example of radar hydrometeor classification. A vertical cut, possibly with lightnings over plotted, would greatly help readers not familiar with radar classification, and may serve as an effective visual introduction to the main topic of the paper.

*We thank the reviewer for the suggestion. We have added an explicit reference to one of the examples in the paper by Besic et al. (2016):*

An example of output can be found in Fig. 11 of the aforementioned paper.

*However, we have not added an extra figure since there are already 20 complex figures in the paper and a hydrometeor classification cut is not strictly relevant for the understanding of the paper. Moreover, we are preparing a new submission focusing on two relevant events observed during the campaign where there will be plots as suggested.*

SPECIFIC COMMENTS AND MINOR CORRECTIONS

- When discussing the hydrometeor type in relation with the lightnings, percentages are given to represent the relative occurrence of given particle types. The first comment I have is that providing these numbers with two decimals accuracy is misleading. We know that hydrometeor classification is subject to many assumptions and uncertainties, so I would recommend providing these percentages with a lower level of accuracy (rounding to integers may be enough in general). The second comment is that there are large portions of the manuscript with a detailed listing of these percentages (e.g. page. 11, lines 3-9) which makes reading a bit hard. I recommend moving as much as possible those numbers to tables and leave in the text only the most relevant ones, focusing on the relevant qualitative behavior.

*We have reduced the accuracy to 1 decimal instead of two. We think we should not reduce it further since there are categories that do not reach 1% but their presence is relevant. We have also created a table summarizing these figures as suggested.*

- P.6, Line 24: "ZPhi" method should read "ZPHI", and the proper reference should be the Testud paper:

Testud, J., E. Le Bouar, E. Obligis, and M. Ali-Mehenni, 2000: The rain profiling algorithm applied to polarimetric weather radar. J. Atmos. Oceanic Technol., 17, 332–356.

Indeed, Ryzhkov used the ZPHI method for rainfall estimation using the same ZPHI method as in Testud et al., 2000 for deriving specific attenuation.

*We changed the naming of the algorithm and added the reference to Testud. We kept the reference to Ryzhkov though, because there are subtle differences in the implementation between the two papers and our approach resembles better the one of Ryzhkov.*

- P. 7, Lines 9-10: according to Besic et al., 2018, if there is a dominant hydrometeor type, the entropy is low (close to 0), while in case of mixtures, the entropy is higher (up to 1). Please explain and/or correct why it is here stated the opposite ("..within the radar resolution volume there is a clearly dominant hydrometeor type (entropy 1) or if it is an heterogeneous mixture without any dominant hydrometeor type (entropy 0)."). Also check that the values of entropy discussed through the paper are consistent with the correct definition.

*We thank the reviewer for having spotted this mistake. We have corrected it and revised the text throughout.*

- P.7, Line 29. "extend" -> "extent"

*We corrected that in the new version*

- P. 9, L25: "The reflectivity data (topmost panels in Fig. 7) of all sources show a bi-modal distribution" Any idea why? Could there be a relation with the bimodality of lightning data?

*We have analyzed data collected on individual days and also plotted 2D histograms of altitude-reflectivity for the entire dataset. What we have observed is that flashes tend to originate at relatively high altitudes (6000-10000 m MSL) and in regions of high reflectivity (30-50 dBZ). When considering all VHF sources, though, several possibilities appear. Most sources are confined to an altitude*

*between 6000-8000 m MSL and a reflectivity of about 40 dBZ. However, two secondary regions appear: one at a similar altitude but with reflectivity 20 dBZ and another with similar reflectivity but at lower altitudes. See the Figures below:*

[Figure]

*We think that the reason for this is that flashes tend to propagate in horizontal layers. Thus, a flash will initiate at the core of a convective cell and then either propagate horizontally at a similar altitude reaching areas out of the core (and therefore with lower reflectivity) or move down to a lower layer within the convective core. We have added this analysis to the text:*

It is worth noticing that a similar behavior was observed when looking at the altitude of the VHF sources (Fig. 6) We think that this is due to the tendency of flashes to preferentially propagate horizontally through layers of high density of charge. The 2D-histogram of altitude-reflectivity and the analysis of individual storms (not shown) show that there are 3 areas with higher density of flashes. Indeed, most sources are concentrated in an area with roughly the same altitude and reflectivity as the flash origin while two other less dense preferential areas appear: One at roughly the same altitude but with lower reflectivity and another with similar reflectivity values but at another altitude. Our interpretation is that flashes are likely to either propagate horizontally at similar altitudes to where they are generated, sometimes extending beyond the convective core (hence the lower reflectivity), or move to a lower layer within the convective core.

- Please check the label for aggregates, sometimes it is referred to as "AG" (P. 6, L. 32), but in fig. captions it is often called "DS".

*We corrected that in the new version.*

- P. 13, Lines 11-14. Here the authors hint at a possible relation between the very high proportion of positive lightnings and hail at the ground. Please add proper references.

*We have added the following lines:*

A higher proportion of +CG flashes have been linked to severe hail-bearing storms in past studies (see the introduction of Pineda et al., 2016 for a summary)

- P. 15, Lines 3-4: "For our classification we have considered as belonging to the mixed phase or liquid regions flashes the first VHF source of which was located in areas where the dominant hydrometeor: : :" I suggest rewording, i.e. "For our classification we have considered as belonging to the mixed

phase or liquid regions flashes whose first VHF source was in areas where the dominant hydrometeor..”

*We changed the sentence accordingly in the revised version.*

- P. 16, Lines 6-7: have you analysed in detail the "origin" of these 2 deg/km Kdp values (lower-right panel in fig. 18)? Such anomaly in the distribution is quite suspect, could it be due to some artifact in the data processing?

*We have decided to add the values out of the range of the presented histograms in the first and last bin respectively. Thus the 2 deg/km value should be read as 2 deg/km or larger (we already say so in the mentioned text) and there is no anomaly in the data processing. We explain that at the beginning of section 3.2:*

It should be noticed that in all the histograms presented in this paper, the values outside of the histogram range are added to the bins at the extremes, e.g., the last bin in the histograms in Fig. 4, top left, include all the values above 900 ms.

*We have also added the following sentence to all the figures showing histograms:*

Note that the values outside of the histogram range are added to the bins at the extremes.

- P.17, Line 2: "flashes with origin in the liquid and mixed phase layers as a proxy for upward lightning". Is there a proper reference to add here? Or is it just a "common sense" expectation?

*This is common sense expectation. We have not found an explicit reference in the literature.*

- P. 17, Lines 20-22: I also suggest considering "riming" (which implies the presence of supercooled water) in addition to particle concentration/size to explain the higher reflectivity.

*We agree with the reviewer and we have modified the sentence as follows:*

The reflectivity at the flash origin location, in particular, has a significantly larger median, suggesting that CG flashes are more likely to occur in regions of higher particle concentration and/or larger particle size and density, increased for example as a consequence of riming.

- Fig. 1. Either add a length scale in the image, or at least mention the size of the domain in the caption. A map with lat, lon axes would be more useful for easier interpretation of fig. 3.

*We have added the scale and lat, lon axes to both images.*

- Fig. 2 Need larger font for the dates.

*We enlarged the font*

- Fig. 4 (and all other similar histograms). Please add a legend with the color meaning (in addition to mentioning in the caption), it is sufficient in the first panel.

*We added a legend to the first panel of each plot.*

- Fig. 8, top panels: it may be better to add the hydrometeor type (e.g. "RN",.. ) directly on the x-axis (so the reader is not forced to keep switching between the number and the label), or at least mark the most relevant hydrometeor types directly on the histogram bars.

*We added the hydrometeor type in the x-axis.*

- Fig. 9: same as above, please mark the hydrometeor type on the axes, try to avoid using the numeric index. Suggestion: consider using a monochromatic color scale (e.g. gray scale), with light tone for low occurrence and dark tone for higher occurrence. I argue this may improve the readability (this is just a suggestion!). The current dark blue really dominates the visual impact.

*We added the hydrometeor type in the axis as suggested. As for the colour scale we think the current colour scale highlights best the most dominant hydrometeors.*

**References**

Pineda, N., Rigo, T., Montanyà, J., and van der Velde, O. A.: Charge structure analysis of a severe hailstorm with predominantly positive cloud-to-ground lightning, Atmospheric Research, 178-179, 31 – 44, 2016

**Response to reviewer 2 of amt-2019-31**

In this document we provide answers to the comments of reviewer 1 of the paper amt-2019-31. Our answers to the reviewer are given in *italic* font. Proposed changes to the manuscript are highlighted in blue color.

The paper presents comprehensive statistical analysis of lightning data associated with concurrent polarimetric radar observations in the Swiss Alps. There is no doubt that the manuscript contains rich information about statistical characteristics if intra-cloud and cloud-to-ground flashes such as their intensity, duration, area, altitude and temperature intervals where flashes originated, etc.

*We thank the reviewer for this positive assessment.*

It comes as no surprise that both IC and CG flashes mostly originate in the areas of dry graupel (called rimed particles in the manuscript) and hail well above the freezing level in sufficiently deep convective clouds.

*Indeed that was expected but we have followed a statistical approach over a relatively large dataset in order to get objective information.*

The authors relate lightning flashes to the output of the MeteoSwiss semi-supervised polarimetric classification algorithm and even estimate the entropy of hydrometeor classification in the flash locations. I am not sure that the use of such a "big gun" as the polarimetric classifier is fully justified in this context. Indeed, polarimetric radar variables such as ZDR, KDP, and hv bear very little classification potential in cold parts of convective storms unless large hail growing in a wet growth regime is observed in the cloud. In fact, discrimination between snow, graupel, and hail aloft is almost exclusively made based on the radar reflectivity factor. Radar reflectivity of hail is larger than the one of graupel and snow.

*As it is described in the papers by Besic et al., our hydrometeor classification is based on the combination of reflectivity, differential reflectivity, specific differential phase, co-polar correlation coefficient and distance to the iso-0°/temperature. The algorithm essentially looks for the minimum distance between the observations and a set of centroids in the measurement space. While it is true that reflectivity is the dominant driver in the classification at sub-zero, the other variables play a non-negligible role in the allocation of the dominant hydrometeor. Moreover, in the liquid and mixed phase layers polarimetry plays an even more relevant role in the classification. Before this particular campaign, our hydrometeor classification has been extensively tested also by comparing the output with in-situ measurements on the ground in mountainous areas with positive results. Therefore we think it is fully justified to use our hydrometeor classification scheme. The centroids used in the processing of the data are summarized in the following table:*

| Hydrometeor type | Zh | Zdr | Kdp | RhoHV | Delta_Z |
|---|---|---|---|---|---|
| AG | 13.5829 | 0.4063 | 0.0497 | 0.9868 | 1330.3 |
| CR | 02.8453 | 0.2457 | 0.0000 | 0.9798 | 0653.8 |
| LR | 07.6597 | 0.2180 | 0.0019 | 0.9799 | -1426.5 |
| RP | 31.6815 | 0.3926 | 0.0828 | 0.9978 | 0535.3 |
| RN | 39.4703 | 1.0734 | 0.4919 | 0.9876 | -1036.3 |
| VI | 04.8267 | -0.5690 | 0.0000 | 0.9691 | 0869.8 |
| WS | 30.8613 | 0.9819 | 0.1998 | 0.9845 | -0066.1 |
| MH | 52.3969 | 2.1094 | 2.4675 | 0.9730 | -1550.2 |
| IH/HDG | 50.6186 | -0.0649 | 0.0946 | 0.9904 | 1179.9 |

Large entropy simply means high variability of Z in a given spatial domain that spans typical range intervals of snow, graupel, and hail.

*We respectfully disagree with the reviewer. As described in the paper by Besic et al. 2018, our concept of entropy is way more sophisticated than that and it refers to the entropy of the observations within the radar range gate. By estimating the entropy at the observations space, we are able to determine whether the composed signal is due to a single hydrometeor type or a combination of different hydrometeors, something that we think it is very relevant for this study.*

I am surprised by the fact that the histograms of ZDR in the regions of lightning initiation above the freezing level are almost perfectly symmetric around 0 dB value. Graupel and hail – major source of lightning flashes – can grow only in sufficiently strong convective updrafts commonly manifested by the ZDR columns. Various researchers report close association of lightning locations and ZDR columns which is not examined and even mentioned in the paper.

*ZDR columns as an indicator of lightning activity are indeed mentioned in the introduction:*

Other authors have observed that the presence of a Zdr column is an indicator of a strong updraft (Snyder et al., 2015), which has been repeatedly reported to favor lightning activity (Calhoun et al., 2013).

*While it is indeed an important mechanism for the generation of lightning activity it is certainly not the only one. As mentioned in the paper:*

The Zdr data (upper-middle panels) exhibit a similar Gaussian-like shape both when all sources are considered and when only the first source is considered. In both cases the distribution is centered around 0 but with very long tails.

*The tails of the distribution are fully compatible with the presence of ZDR columns in the data. However, ZDR columns are typically observable minutes before the intensification of a storm (see the paper by Snyder et al 2015) and from our data we can conclude that on average more lightning is produced at more mature phases of the storm when particles have grown into a more spherical shape.*

At the same time, there is apparent sign of nozero positive KDP indicated in the histograms in Figs. 7 and 12. What is the origin of these positive values of KDP in cold parts of convective clouds?

Horizontally oriented ice crystals in the proximity of graupel and hail or the tops of KDP columns? In the latter situation, ZDR columns with noticeably positive ZDR should be also observed.

*It is a well-known feature that large ice crystals (or not particularly large but rather oblate such as dendrites, i.e. Bechini et al. 2013), rimed particles and hail may have non-zero values of KDP. Simple T-matrix calculations already show this positive KDP values. For illustration, we show here relationships between Zh and ZDR for various hydrometeors computed using T-matrix simulations:*

[Figure]

a)

b)

c)

d)

**Figure 1 Zh-KDP relationship for different hydrometeors: a) ice crystals, b) wet snow, c) graupel, d) hail**

GENERAL COMMENT

The present study aims at combining LMA and polarimetric radar observations to infer relationships between lightning activity and cloud microphysics. It is based on a significant dataset collected in the northeastern part of Switzerland over a period of two and a half months during summer 2017. Overall this study provides interesting findings regarding potential interactions between the microphysical cloud structure and lightning initiation/propagation.

*We thank the reviewer for his overall positive assessment.*

It is, however, extremely descriptive and definitely lacks some physical interpretation to support the results. Section 3, for instance, mostly consists in a discussion of endless series of numbers and percentages / histograms with no interpretation and no comparison against previous findings.

*We have added a subsection (Discussion) in Section 3 to put our results into context.*

Authors also lack perspective on the performance of their classification algorithm, especially as we all know that dual-polarimetric HCA are subject to large uncertainties (low percentage values such as discussed in the paper don't really make sense and should definitely be avoided). Actually I believe that Section 3 should be simplified and rewritten to highlight the most significant results as it is very difficult to comment on the results as they stand.

*We respectfully disagree with the reviewer. Our classification algorithm has been thoroughly tested in multiple scenarios with very good results. As for the low percentage values presented in the paper, we think they are still relevant to showcase that very few lightning initiates in such conditions regardless of the specific percentage. We do agree that a precision up to 2 digits after the comma may be misleading with respect to the accuracy of the classification and, therefore, in the revised version we rounded the values to just one.*

*In the revised version we also moved most of the numbers into tables in order to improve the readability of the manuscript and we provided more physical context to the measurements.*

SPECIFIC COMMENTS

More details are needed regarding the Hydrometeor Classification Algorithm used in this study. Authors should also discuss the uncertainty associated with their method, especially regarding hail/graupel identification. Actually I am quite surprised by the high proportion of lightning initiation and propagation that takes place in hail. Most past studies have found that graupel is by far (70-90%) the preferred environment for lighting initiation. Also, author are using a C-band radar. However we

know that at such frequency hail identification might be a problem due to resonance scattering effects. Please comment.

*We believe that explaining the hydrometeor classification in detail would unnecessarily lengthen the manuscript. The algorithm has been thoroughly described in the two papers by Besic et al. and a considerable amount of work has been put in the validation of the algorithm in the past. The two papers are published in open access journals so the interested reader should not have any difficulty to access them.*

*The higher than reported percentage of lightning initiating in hail areas on the other hand may be due to the fact that in our dataset hail-producing convective cells may indeed be over-represented. A significant amount of lightning data was gathered on days (notably August the 1$^{st}$) when hail was observed on the ground. We have added the following lines at the beginning of Section 3.4.1:*

In any case, the proportion of +CG flashes with respect to the total number of CG flashes (41%) is significantly higher than that observed on the Säntis tower over a 2-year period (15%) (Romero et al., 2013). That is due to the fact that on three out of the 8 analyzed days (10 and 19 July and 1 August), the proportion of +CG flashes is abnormally high (see Table 2). The percentage of +CG the 19 July (72.6%) is particularly noteworthy. On that day, large swathes of terrain south of the Säntis tower were affected by hail according to the POH algorithm. There was also extensive hail recorded on August 1. A higher proportion of +CG flashes have been linked to severe hail-bearing storms in past studies (see the introduction of Pineda et al. (2016) for a summary)

*As for the detection of hail, resonance effects may be an issue for hail quantification in the C-band (although using other parameters such as KDP partially mitigates that issue) but we do not see it as an issue for the actual hail detection.*

Radar scientists claim to be able to identify as many hydrometeor types as possible. However, due to the high level of uncertainty in HID, it would be more realistic to use less categories. This would also ease the interpretation of the results.

*We respectfully disagree with the reviewer. Based on the centroids on which our classification algorithm is based, we think that the current categories correspond to perfectly separable data and we have reduced the hydrometeor types to the bare minimum useful for microphysics studies.*

As mentioned previously, this paper lacks physical interpretation of the results. Results should be compared with the literature (especially regarding hail) and more details should be given about the role of the microphysical environment in the initiation/propagation of lightning. Also what is the influence of orography on lightning? As mentioned by the authors, the particularity of this study is that it takes place over very complex terrain. Hence the effect of mountains should be considered when analysing the results.

*The effect of orography on lightning are not explicitly discussed in the paper and it is a complex issue that requires further analysis that it is out of the scope of this paper. Nevertheless, some relevant aspects regarding the orography are shown in the paper such as the higher density of lightning in a narrow region at the footsteps of the Alps and the presence of hot spots of CG lightning activity visible in Fig. 14 in the paper at the location of some mountains.*

The spatial extension of the domain of analysis seems rather small to me. LMA data can potentially be used up to 100/150 km without any problem. Please comment.

*LMA networks typically consist of about 10 sensors or more whereas the LMA deployed had only between 5 and 6 sensors operating at any time which is the minimum to have reliable data. Moreover, as it is explained in Section 2.1.2, for practical reasons, the sensors were placed in the vicinity of mobile phone base stations which resulted in an increased ambient noise and, therefore, a decrease in sensitivity. Furthermore, the presence of the Alps resulted in the network being essentially blind south of its location. Nevertheless, there was indeed some level of detectability up to 100 km. If you examine Fig. 1, there are actually two domains presented. The extended domain, where some lightning activity was detected during the campaign, corresponds roughly to a 100-km diameter area. The reduced domain used is a conservative approach that ensures a high level of detectability even when only 5 VHF sensors were operational.*

I do not see the need for 20 figures. Most of the results shown in Figs. 4 to 20 could be summarized in a few tables. Instead, authors should include the analysis of two contrasting events in their study and discuss the storm structures, associated horizontal/vertical cross-sections of reflectivity / polarimetric moments, and location of LMA sources within the storms.

*The initial intention when writing the article was to show as much as possible the actual data used so that readers can also draw their own conclusions. We have already up to a certain extent reduced the number of figures presented. As requested by the reviewers, in the revised version we provide more physical interpretations of the results and relate them with past literature but we still think that the figures provided are relevant and useful.*

*As for the case studies, we plan to submit two other papers related to the campaign. One will deal with evidence of upward lightning produced by the Säntis tower while the other indeed focuses on two contrasting convective events. We think that the generalist statistical approach followed in this particular study is sufficiently worthy to be published.*

A detailed comparison between EUCLID and LMA detection capability would be interesting.

*That would be indeed interesting but we think it is out of the scope of this particular study. The EUCLID detection capability in the vicinity of the Säntis tower has been examined in the past by the authors (e.g. Azadifar et al., 2015). As for the LMA detectability, we are confident that it does not play a role in the presented data since most of the flashes analysed were detected in the reduced domain where we had high detectability.*

*The detection capabilities of LMA networks in similar rough terrain have been analysed in the framework of the HyMex campaign (Pédeboy et al, 2014) and with the LMA installed in Corsica (Pédeboy et al., 2018).*

A discrimination between convective and stratiform regions of the storms would also help interpreting the results.

*The discrimination between convective and stratiform regions poses several problems since these are ill-defined categories and never so clear cut. Our approach is radar bin-based, rather than radar region-based. We think there is a certain originality and usefulness in our approach based on*

*assessing the meteorological conditions from a radar point of view at the location of lightning initiation and through its propagation path that make it worth publishing as it is.*

**References**

Pédeboy S., Defer E., Schulz W.: Performance of the EUCLID network in cloud lightning detection in the South-East France. 8[th] HyMeX Workshop, 15-18 Sept. 2014, Valetta, Malta

Pédeboy S., Barnéoud P., Defer E., Coquillat S.: Analysis of the Intra-Cloud lightning activity detected with Low Frequency Lightning Locating Systems. 25[th] International Lightning Detection Conference & 7[th] International Lightning Meteorology Conference, 12-15 March 2018, Ft. Lauderdale, Florida, USA

[revised manuscript text omitted]
, 55.09can be found in areas with rimed particles, 25.89in areas with solid hail, 11.96in dry snow and 1.85in ice crystals areas (of which 70.05are vertically oriented) , 1.76in wet snow, 1.50in rain (of which 24.28in light rain) and 0.32in melting hail. 1.62propagate in areas where no radar echo was detected. When looking only at the first sources, 54.85are produced in areas with rimed particles, 28.38in areas with solid hail, 9.49in dry snow, 2.05in ice crystals areas (of which 62.61are vertically oriented), 0.79in wet snow, 0.90in rain (37.76of which in light rain) and 0.24in melting hail. 3.29originate 
[revised manuscript text omitted]